# AdaFM: Adaptive Variance-Reduced Algorithm for Stochastic Minimax Optimization

## Abstract

In stochastic minimax optimization, variance-reduction techniques have been widely developed to mitigate the inherent variances introduced by stochastic gradients. Most of these techniques employ carefully designed estimators and learning rates, successfully reducing variance. Although these approaches achieve optimal theoretical convergence rates, they require the careful selection of numerous hyperparameters, which heavily depend on problem-dependent parameters. This complexity makes them difficult to implement in practical model training. To address this, our paper introduces Adaptive Filtered Momentum (AdaFM), an adaptive variance-reduced algorithm for stochastic minimax optimization. AdaFM adaptively adjusts hyperparameters based solely on historical estimator information, eliminating the need for manual parameter tuning. Theoretical results show that AdaFM can achieve a near-optimal sample complexity of $O(\epsilon^{-3})$ to find an $\epsilon$-stationary point in non-convex-strongly-concave and non-convex-Polyak-Łojasiewicz objectives, matching the performance of the best existing non-parameter-free algorithms. Extensive experiments across various applications validate the effectiveness and robustness of AdaFM.

## 1 Introduction

Typically, the stochastic minimax optimization problem Nouiehed et al. (2019); Lin et al. (2020); Lu et al. (2020); Huang et al. (2022; 2023) can be formulated as follows:

$$\min_{x \in \mathbb{R}^{d_1}} \max_{y \in \mathcal{Y}} f(x, y) = \mathbb{E}_{\xi \in \mathcal{D}}[f(x, y, \xi)], \tag{1}$$

where data sample $\xi$ is a random variable following an unknown distribution $\mathcal{D}$. $\mathcal{Y} \subset \mathbb{R}^{d_2}$ is closed and convex, and $f : \mathbb{R}^{d_1} \times \mathbb{R}^{d_2} \to \mathbb{R}$ is non-convex in $x$. We call $x$ the primal variable and $y$ the dual variable. Problem in equation 1 is widely used in many machine learning applications, e.g., adversarial training Goodfellow et al. (2014b); Miller et al. (2020), Generative Adversarial Network (GAN) Arjovsky et al. (2017); Goodfellow et al. (2014a), deep Area Under the Curve (AUC) Yuan et al. (2021; 2022), and sharpness-aware minimization Foret et al. (2021); Qu et al. (2022).

Since stochastic gradients on both the primal and dual parameters inherently exhibit variance Johnson & Zhang (2013); Dubey et al. (2016), which slows down the convergence rate, recent studies have focused on Variance-Reduction (VR) techniques Reddi et al. (2016); Xu et al. (2017); Cutkosky & Orabona (2019); Ward et al. (2020); Huang et al. (2022); Xu et al. (2023); Huang et al. (2023); Liu et al. (2023) to mitigate this variance, demonstrating the ability to achieve optimal sample complexity of $O(\epsilon^{-3})$ for finding an $\epsilon$-stationary point.

While the aforementioned VR-based algorithms have proven highly successful at the theoretical level, they often perform poorly in actual model training Defazio & Bottou (2019); Arjevani (2017). One significant reason is that VR techniques introduce numerous hyperparameters that must be carefully selected in minimax optimization to ensure the effectiveness of the VR techniques. For instance, in stochastic minimax optimization, the VR-based algorithms mentioned above do not always guarantee convergence if the ratio of the learning rates for $x$ and $y$ is not selected appropriately Yang et al. (2022a). Moreover, the large number of hyperparameters makes the algorithm highly sensitive, such that even small hyperparameter changes can prevent the algorithm from converging.

To verify the above issues, we conducted real model training, specifically using WGAN-GP Gulrajani et al. (2017), on CIFAR10 and CIFAR100 Krizhevsky et al. (2009). From Figure 1, we observe that

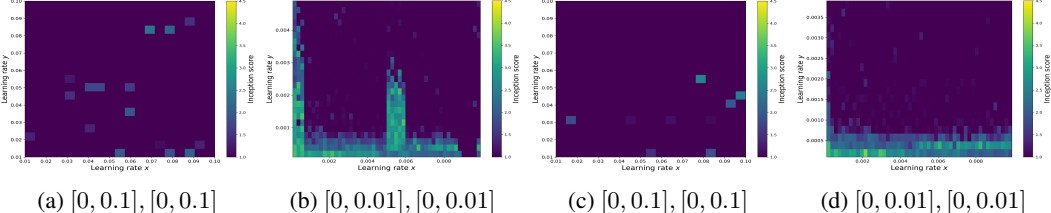

(a) $[0, 0.1], [0, 0.1]$     (b) $[0, 0.01], [0, 0.01]$     (c) $[0, 0.1], [0, 0.1]$     (d) $[0, 0.01], [0, 0.01]$

Figure 1: The hyperparameter grid search of RSGDA on CIFAR10 and CIFAR100. Figures 1a and 1b display the results of the search on CIFAR10 using two different hyperparameter grids. Similarly, Figures 1c and 1d show the results on CIFAR100. The grid search was performed in the range $[0, 0.1]$ with a step size of 0.005 and in the range $[0, 0.01]$ with a step size of 0.0002.

RSGDA faces several challenges. First, when we select the parameters from a large space, i.e., $[0, 0.1]$ of the two learning rates in Figures 1a and 1c, we can see that most results are not desired enough. As such, we need to compress the searching space. Consequently, the parameters are highly sensitive; for example, as shown in Figure 1b, when the learning rate of $x$ is very small (i.e., less than 0.002), even a slight change in the learning rate of $y$ can directly prevent the algorithm from functioning properly, particularly when the learning rate of $y$ is around 0.002 or 0.001. Lastly, changes in the dataset cause the space of effective parameters to shift dramatically, making it difficult to provide a default combination of parameters for different datasets and tasks. This results in a highly computationally laborious hyperparameter search for various tasks. Therefore, to enhance the practicality of VR-based algorithms, it is necessary to address the issue of excessive hyperparameters.

In minimization problems, the parameter-free approach Kingma & Ba (2014); Li & Orabona (2019); Ward et al. (2020); Levy et al. (2021) offers an intuitive solution to enhance VR-based algorithms by automatically adapting hyperparameters, thus avoiding manual tuning. However, implementing VR techniques in a parameter-free manner for minimax problems remains highly challenging because minimax problems require the simultaneous consideration of updates to both variables. As a result, traditional VR-based algorithms for minimax problems involve nearly twice as many hyperparameters compared to those used in minimization problems. specifically, VR techniques maintain gradient estimators $v_t$ and $w_t$ for $x$ and $y$, respectively, with the corresponding learning rates $\eta_x^t$ and $\eta_y^t$ carefully designed based on $v_t$ and $w_t$. Many hyperparameters in $v_t$, $w_t$, $\eta_x^t$, and $\eta_y^t$ require knowledge of problem-dependent parameters to be chosen properly, ensuring the effectiveness of VR-based algorithms. These problem-dependent parameters, such as the smoothness constant $L$ and the gradient bound $G$, are difficult to determine during actual model training. This raises a natural question:

> **Can we design an adaptive VR-based algorithm to achieve the optimal convergence rate in the minimax optimization problem?**

In this paper, we introduce an adaptive VR-based algorithm named Adaptive Filtered Momentum (AdaFM) for stochastic minimax optimization problems. Inspired by STORM Cutkosky & Orabona (2019), AdaFM incorporates variance reduction with momentum correction and features a novel update method for both momentum parameters and learning rates, making them adaptive and simplifying their computation, thus enhancing ease of use. Specifically, The momentum parameter only decreases with the number of iterations, thus avoiding parameter tuning and improving the stability of the algorithm. The learning rate takes multiple factors into full account. On one hand, the learning rate decreases as the cumulative value of the estimator increases. On the other hand, the learning rates of $x$ and $y$ interact with each other, ensuring that the step sizes of $x$ and $y$ adapt to the desired ratio. The main contributions of this paper are summarized as follows:

- We introduce AdaFM, the first adaptive VR-based algorithm for stochastic minimax optimizations. AdaFM is an adaptive method that achieves the near optimal convergence rate in the minimax optimization problem. AdaFM dynamically adjusts the momentum parameters according to the number of iterations and automatically adjusts the learning rate based on the current momentum parameters and historical estimator information.

- We provide detailed analyses of AdaFM in both Non-Convex-Strongly-Concave (NC-SC) and Non-Convex-Polyak-Łojasiewicz minimax (NC-PL) settings. Although the theoretical

result in the NC-PL setting is worse than NC-SC due to the more complicated property, both of them can achieve an $\epsilon$-stationary point with an optimal complexity of $O(\epsilon^{-3})$ in short. They match the best result among existing VR-based parametric algorithms.

- We evaluate our AdaFM across various learning tasks formulated by the stochastic minimax optimization, including (1) two distinct test functions, (2) deep AUC Yuan et al. (2021) on an NC-SC objective, and (3) training Wasserstein-GANs Arjovsky et al. (2017) to validate the NC-PL objective. Experimental results indicate that AdaFM exhibits greater robustness than other traditional parametric VR-based algorithms and consistently outperforms TiAda.

## 2  RELATED WORK

**Stochastic Minimax Optimization.** Stochastic minimax optimization has gained significant traction in various machine learning applications. The prevailing approach for solving minimax optimization problems typically involves alternating between optimizing the minimization and maximization sub-problems, which are typically addressed by stochastic gradient descent ascent (SGDA) Nouiehed et al. (2019); Lin et al. (2020); Lu et al. (2020). Notably, they can achieve a sample complexity of $O(\epsilon^{-4})$ in stochastic settings Nouiehed et al. (2019); Lin et al. (2020); Yang et al. (2020). Subsequently, some accelerated algorithms utilizing adaptive learning rates have been extended to minimax optimization, both theoretically and practically. These include approaches for strongly-convex strongly-concave problems Antonakopoulos et al. (2021), nonconvex-convex problems Yang et al. (2022a); Huang et al. (2023), and nonconvex-PL problems Huang (2023); Guo et al. (2023). For example, Guo et al. (2023) proposes PES to address the primal objective and duality gaps under the NC-PL setting.

**VR Techniques.** VR techniques have gained prominence in stochastic optimization, addressing the inherent variance issue associated with stochastic gradients. Notable approaches include stochastic variance reduced gradient Johnson & Zhang (2013); Reddi et al. (2016), SPIDER Fang et al. (2018); Li et al. (2023b), and STORM Cutkosky & Orabona (2019); Levy et al. (2021), which have accelerated the convergence. SPIDER has led to the development of fast HAPG Shen et al. (2019) and SVRPG Xu et al. (2020b). Momentum-based techniques such as ProxHSPGA Pham et al. (2020), SVMR Jiang et al. (2022), and NSTORM Liu et al. (2023) have emerged from STORM's principles, addressing various optimization scenarios. While VR-based algorithms have demonstrated efficient convergence results, the challenge of reducing the search space for hyper-parameters remains under-explored.

**Parameter-Free Algorithms.** Parameter-free algorithms have significantly enhanced their utility by adapting to various parameters without the need for extensive manual tuning. Some adaptive optimizers that achieve this property include AdaGrad Duchi et al. (2011), Adam Reddi et al. (2016), and STORM+ Levy et al. (2021). TiAda Li et al. (2023a) extends this adaptivity to minimax optimizations by separating the two timescales. Additionally, parameter-free algorithms have been extensively developed in online learning. For instance, Beygelzimer et al. (2015) focuses on online boosting, Xu et al. (2020a) addresses online reinforcement learning, and Hanneke et al. (2023) explores multi-class online learning. In these contexts, the primary goal is for the learner to compete with the performance of the best possible function $f$, thereby achieving minimal regret. Note that online learning primarily addresses the cold data streaming problem, which is parallel to this paper.

## 3  THE PROPOSED ALGORITHM

To achieve the adaptive method, we introduce the parameter-free algorithm, called Adaptive Filtered Momentum (AdaFM), to solve the minimax optimization problem in equation 1, which is illustrated in Algorithm 1. Specifically, we leverage similar VR estimators for the primal variable $x$ and the dual variable $y$, denoted as $v_t$ and $w_t$ inspired by STORM Cutkosky & Orabona (2019). In each iteration $t$, the two estimators $v_t$ and $w_t$ can be calculated as follows:

$$v_t = \nabla_x f(x_t, y_t; \xi_t^x) + (1 - \beta_t)(v_{t-1} - \nabla_x f(x_{t-1}, y_{t-1}; \xi_t^x)), \qquad (2)$$

$$w_t = \nabla_y f(x_t, y_t; \xi_t^y) + (1 - \beta_t)(w_{t-1} - \nabla_y f(x_{t-1}, y_{t-1}; \xi_t^y)). \qquad (3)$$

However, if the original momentum parameter update method is directly used, it has been proven by Huang et al. (2023); Huang & Gao (2023); Liu et al. (2023) that designing different momentum parameters $\beta_t^x$ and $\beta_t^y$ for the two variables $x$ and $y$ is required. This inevitably introduces more additional hyperparameters. To address this problem, we simplify the momentum parameters for both

---

**Algorithm 1** Learning procedure of AdaFM.

---

**Initialization:** $(x_1, y_1)$, $\gamma, \lambda > 0$, $\frac{1}{3} > \delta > 0$;
1: **for** $t = 1$ to $T$ **do**
2:     sample $\xi_t^x$ and $\xi_t^y$;
3:     **if** $t = 1$ **then**
4:         $v_t = \nabla_x f(x_t, y_t; \xi_t^x)$, $w_t = \nabla_y f(x_t, y_t; \xi_t^y)$;
5:     **else**
6:         Update the estimators $v_t$ and $w_t$ via equation 2-equation 3;
7:     **end if**
8:     Update the momentum parameter $\beta_{t+1} = 1/t^{2/3}$;
9:     Update $\alpha_t^x$ and $\alpha_t^y$ via equation 5;
10:    Update learning rates $\eta_t^x$ and $\eta_t^y$ via equation 4;
11:    $x_{t+1} = x_t - \eta_t^x v_t$,   $y_{t+1} = \mathcal{P}_{\mathcal{Y}}(y_t + \eta_t^y w_t)$
12: **end for**

---

variables by setting $\beta_{t+1} = 1/t^{2/3}$. This means that $\beta_t$ only changes with the number of iterations, making it tuning-free. Such a simplification is made possible by our careful design of the learning rates. Below, we describe how to update the learning rates $\eta_t^x$ and $\eta_t^y$ for the two variables:

$$\eta_t^x = \frac{\gamma}{\max\{\alpha_t^x, \alpha_t^y\}^{1/3+\delta}}, \quad \eta_t^y = \frac{\lambda}{(\alpha_t^y)^{1/3-\delta}}, \tag{4}$$

where

$$\alpha_t^x = \sum_{i=1}^t \frac{\|v_i\|^2}{\beta_{i+1}}, \quad \alpha_t^y = \sum_{i=1}^t \frac{\|w_i\|^2}{\beta_{i+1}}. \tag{5}$$

It seems that there are three extra hyperparameters appearing in learning rates $\eta_t^x$ and $\eta_t^y$ in equation 4: $\gamma$, $\lambda$, and $\delta$, require manual tuning. In particular, we will delve into these hyperparameters later and demonstrate that convergence can be achieved even without manual adjustments. Now we explain why we choose the momentum parameters and learning rates this way. Our choices are inspired by the analysis of dynamic errors in both variables, denoted as $\epsilon_t^x := v_t - \nabla_x f(x_t, y_t)$ and $\epsilon_t^y := w_t - \nabla_y f(x_t, y_t)$. Dynamic error reflects the error between the current estimator and the true gradient on each iteration $t$. More specifically, based on the update rule of $v_t$ and $w_t$ in our proposed AdaFM algorithm, the error dynamics can be obtained as follows:

$$\epsilon_t^x = (1 - \beta_t)\epsilon_{t-1}^x + (1 - \beta_t)Z_t^x + \beta_t(\nabla_x f(x_t, y_t; \xi_t^x) - \nabla_x f(x_t, y_t)), \tag{6}$$

$$\epsilon_t^y = (1 - \beta_t)\epsilon_{t-1}^y + (1 - \beta_t)Z_t^y + \beta_t(\nabla_y f(x_t, y_t; \xi_t^y) - \nabla_y f(x_t, y_t)), \tag{7}$$

where

$$Z_t^x = (\nabla_x f(x_t, y_t; \xi_t^x) - \nabla_x f(x_{t-1}, y_{t-1}; \xi_t^x)) - (\nabla_x f(x_t, y_t) - \nabla_x f(x_{t-1}, y_{t-1})),$$

$$Z_t^y = (\nabla_y f(x_t, y_t; \xi_t^y) - \nabla_y f(x_{t-1}, y_{t-1}; \xi_t^y)) - (\nabla_y f(x_t, y_t) - \nabla_y f(x_{t-1}, y_{t-1})).$$

The third term on the RHS of equation 6-equation 7, namely $\nabla_x f(x_t, y_t; \xi_t^x) - \nabla_x f(x_t, y_t)$ and $\nabla_y f(x_t, y_t; \xi_t^y) - \nabla_y f(x_t, y_t)$, represents the error between the stochastic gradient and the true gradient. This error is generally controlled by choosing a decreasing value for the momentum parameter $\beta_t$. For instance, in STORM Cutkosky & Orabona (2019), the momentum parameter is defined as $\beta_{t+1} = c\eta_t^2$, where $\eta_t = \theta/(w + t)^{1/3}$. However, the three hyperparameters $\theta$, $w$, and $c$, which are linked to $L$ and $G$, necessitate configurations that are dictated by problem-dependent parameters. To fulfill our objectives, we streamlined the momentum parameters, setting $\beta_{t+1} = 1/t^{2/3}$ for both $x$ and $y$. As iterations increase, the momentum parameter gradually approaches zero. This ensures that in early iterations, it remains large enough to leverage the acceleration effect of momentum, while in later iterations, it decreases, dissipating the accumulated "momentum potential energy." As a result, the algorithm transitions to Simple SGD, allowing it to converge near the stationary point.

Then, we prepare the choice of learning rate $\eta_t^x$ and $\eta_t^y$ to afford the parameter-free manner. For the error dynamics, while we have addressed the last terms $\nabla_x f(x_t, y_t; \xi_t^x) - \nabla_x f(x_t, y_t)$ and $\nabla_y f(x_t, y_t; \xi_t^y) - \nabla_y f(x_t, y_t)$, there are still elements $Z_t^x$ and $Z_t^y$ that require attention. These

elements reflect the differences in model weights before and after each update. Our analysis suggests that $Z_t^x$ and $Z_t^y$ can be upper-bounded as follows: $\|Z_t^x\|^2 \leq 8L^2((\eta_{t-1}^x)^2\|v_{t-1}\|^2 + (\eta_{t-1}^y)^2\|w_{t-1}\|^2)$ and $\|Z_t^y\|^2 \leq 8L^2((\eta_{t-1}^x)^2\|v_{t-1}\|^2 + (\eta_{t-1}^y)^2\|w_{t-1}\|^2)$. It is worth noting that these bounds are closely related to the learning rates with the smooth property of the functions. Therefore, in order to achieve adaptivity and at the same time fulfill the above requirements, a natural idea is to relate the learning rates to historical estimators' information. Inspired by Adagrad Duchi et al. (2011), we let the learning rates decrease as the historical estimators values accumulate, that is $\eta_t^x = O(1/\sum_{i=1}^t \|v_i\|^2)^{1/3+\delta}$ and $\eta_t^y = O(1/\sum_{i=1}^t \|w_i\|^2)^{1/3-\delta}$, where $\delta$ is an arbitrarily small value.

However, relying solely on historical estimator information makes it difficult to ensure a strictly monotonically decreasing learning rate due to the inherent variance of stochastic gradients. This assurance is crucial. For instance, when the algorithm approaches a stationary point after only a few iterations, the cumulative estimator values $\sum_{i=1}^t \|v_i\|^2$ are still quite small, which can lead to a high learning rate $\eta_t^x$ that is hard to reduce further. This can easily result in oscillations near the stationary point, making it difficult to achieve stability accurately. Therefore, we combine the learning rate with the momentum parameter to ensure a strictly monotonic decrease in the learning rate. Specifically, we define $\eta_t^x = O(\frac{1}{\sum_{i=1}^t \|v_i\|^2/\beta_{i+1}})^{1/3+\delta}$ and $\eta_t^y = O(\frac{1}{\sum_{i=1}^t \|w_i\|^2/\beta_{i+1}})^{1/3-\delta}$.

Moreover, minimax optimizations bring additional challenges in determining the learning rates for both variables. A consensus Lin et al. (2020); Li et al. (2023a) suggests updating $y$ at a higher learning rate than $x$ to ensure that $y$ reaches optimal first. Therefore, $x$ should be updated cautiously if the inner maximization sub-problem is unresolved. Based on these principles, it becomes clear that discussing the learning rates of $x$ and $y$ separately is insufficient. Consequently, when updating $x$, we also consider the learning rate of $y$ by setting $\eta_t^x = O(1/\max\{\alpha_t^x, \alpha_t^y\})^{1/3+\delta}$. This ensures that if the inner maximization sub-problem has not yet been accurately solved, the update of $x$ is always slowed. The final strategy is shown in equation 4. Through this method, we use only information about the number of iterations and the cumulative estimator values to achieve adaptive learning rates.

Finally, we discuss the three parameters: $\gamma$, $\lambda$, and $\delta$. The purpose of $\gamma$ and $\lambda$ is to enable AdaFM to adapt more quickly to various application scenarios. In our proof, we will show that even if we simply set $\gamma = \lambda = 1$, our theorems still hold. Regarding $\delta$, it reflects the degree of scale adjustment of the learning rates for $x$ and $y$. In our proof, we demonstrate that in complex settings, where $\delta$ takes an arbitrarily small value, we can ensure that the convergence rate remains close to $O(T^{-1/3})$, as explained in the next section. Therefore, adjusting these three parameters presents no difficulty, which is consistent with our claim that AdaFM is adaptive.

## 4 THEORETICAL ANALYSIS

In this section, we present the convergence result and sample complexity of our AdaFM algorithm under Non-Convex-Strongly-Concave (NC-SC) and Non-Convex-Polyak-Łojasiewicz (NC-PL) objectives, respectively. We define $(x, y)$ as an $\epsilon$-stationary point if both $\mathbb{E}\|\nabla_x f(x, y)\| \leq \epsilon$ and $\mathbb{E}\|\nabla_y f(x, y)\| \leq \epsilon$, where the expectation accounts for all algorithmic randomness. As shown in Yang et al. (2022a;b); Huang et al. (2023); Huang & Gao (2023); Xu et al. (2023), this definition of stationary can be conveniently translated to the near-stationary of the primal function $\Phi(x) = \max_{y \in \mathcal{Y}} f(x, y)$. Before presenting the theoretical results, we set $\delta_x = 1/3 + \delta$ and $\delta_y = 1/3 - \delta$ to simplify the notation in the following sections. We then state some useful assumptions to facilitate our analysis.

**Assumption 1** (Smoothness). *There exists a constant $L > 0$, such that*

$$\|\nabla f(x_1, y_1; \xi) - \nabla f(x_2, y_2; \xi)\| \leq L\|(x_1, y_1) - (x_2, y_2)\|,$$

*where $x_1, x_2 \in \mathbb{R}^{d_1}$ and $y_1, y_2 \in \mathcal{Y}$.*

**Assumption 2** (Bounded Gradient). *For any $x \in \mathbb{R}^{d_1}$ and $y \in \mathcal{Y}$, there exists a constant $G$ such that*

$$\|\nabla_x F(x, y; \xi^x)\| \leq G \quad and \quad \|\nabla_y F(x, y; \xi^y)\| \leq G.$$

It is worth noting that the problem-dependent in these assumptions are only presented to facilitate our proof; we do not need the information from these assumptions for the implementation of the algorithm. In equation 1, we represent $y^*(x) := \arg\max_{y \in \mathcal{Y}} f(x, y)$ as the solution of the inner maximization

sub-problem. We use $\mathcal{P}_{\mathcal{Y}}(\cdot)$ as projection operator onto set $\mathcal{Y}$. $\kappa = L/\mu$ is the condition number. In addition, we aim to find a near-stationary point for the minimax problem. Accordingly, we introduce an additional assumption as follows:

**Assumption 3.** *(Bounded Primal Function Value) There exists a constant $\Phi_*$ such that for any $x \in \mathbb{R}^{d_1}$, $\Phi(x)$ is upper bounded by $\Phi_*$.*

**Remark 1.** Assumptions 1-2 are used in numerous studies involving adaptive algorithms and minimax optimizations such as Carmon et al. (2019); Yang et al. (2020); Levy et al. (2021); Kavis et al. (2022); Huang et al. (2023); Liu et al. (2023). Particularly noteworthy is Assumption 3, which signifies the bounded nature of the domain of $y$-a condition also considered in AdaGrad Levy (2017); Levy et al. (2018). In neural networks featuring rectified activations, the scale-invariance property Dinh et al. (2017) renders the imposition of boundedness on $y$ compatible with expressive modeling. Additionally, Wasserstein GANs Arjovsky et al. (2017) utilize critic projections to confine weights within a small cube centered around the origin.

### 4.1 ANALYSIS OF THE NC-SC SETTING

We use the following assumption to show the strong concavity property in the dual parameter $y$.

**Assumption 4** (Strongly Concave in $y$). *Function $f(x,y)$ is $\mu$-strongly-concave ($\mu > 0$) in $y$, that is, for any $x \in \mathbb{R}^{d_1}$ and $y_1, y_2 \in \mathcal{Y}$, we have*

$$f(x, y_1) \geq f(x, y_2) + \langle \nabla_y f(x, y_1), y_1 - y_2 \rangle + \frac{\mu}{2} \|y_1 - y_2\|^2.$$

**Theorem 1** (Convergence, NC-SC). *Under Assumptions 1-4, after $T$ training epochs, AdaFM in Algorithm 1 satisfies*

$$\sum_{t=1}^{T} \|\nabla_x f(x_t, y_t)\|^2 + \sum_{t=1}^{T} \|\nabla_y f(x_t, y_t)\|^2 = O\left( \kappa^{2 + \frac{5 + 5\delta_y}{3\delta_y}} T^{\frac{1 - 2\delta_y}{3\delta_y}} + \kappa^{\frac{3}{1 - \delta_x}} T^{\frac{2\delta_x}{3(1 - \delta_x)}} \right).$$

*Then according to the setting of $\delta_x$ and $\delta_y$, we can get*

$$\frac{1}{T} \left[ \mathbb{E} \sum_{t=1}^{T} \|\nabla_x f(x_t, y_t)\| + \mathbb{E} \sum_{t=1}^{T} \|\nabla_y f(x_t, y_t)\| \right] = O\left( \frac{\kappa^{4.5}}{T^{1/3 + \delta}} \right).$$

Our proof of the NC-SC setting can be categorized into four cases based on the magnitude of the cumulative error terms, $\mathbb{E} \sum_{t=1}^{T} \|\epsilon_t^x\|^2$ and $\mathbb{E} \sum_{t=1}^{T} \|\epsilon_t^y\|^2$, as well as the cumulative value of the gradients, $\mathbb{E} \sum_{t=1}^{T} \|\nabla_x f(x_t, y_t)\|^2$ and $\mathbb{E} \sum_{t=1}^{T} \|\nabla_y f(x_t, y_t)\|^2$. When the cumulative error term is relatively large, it acts as an upper bound for the cumulative gradient. However, when the accumulated error term is small, we may not establish an upper bound for the cumulative gradient based solely on the error term. In these situations, we can provide additional information to determine the upper bound for the cumulative gradient.

**Remark 2.** If we aim to achieve the $\epsilon$-stationary point by AdaFM in the NC-SC setting, under the setting that $\delta$ is close to 0, the total number of training epochs should satisfy that the iteration $T$ is arbitrarily close $O(\epsilon^{-3})$. In addition, because AdaFM only needs two samples, i.e., $O(1)$, to compute estimators and gradients in each training epoch, the total sample complexity can arbitrarily achieve $O(\epsilon^{-3})$. According to the analysis, Theorem 1 also holds by simply setting both $\gamma$ and $\lambda$ to 1. It is worth noting that the sample complexity of AdaFM is infinitely close to the optimal sample complexity of parametric algorithms Luo et al. (2020); Huang & Gao (2023); Huang et al. (2023); Xu et al. (2023) in stochastic minimax optimizations. In contrast, as far as we know, Tiada Li et al. (2023a), the only remaining parameter-free algorithm in minimax optimization based on SGDA Nouiehed et al. (2019); Lin et al. (2020), can only achieve the near sample complexity of $O(\epsilon^{-4})$, which is worse than our proposed AdaFM algorithm.

**Remark 3.** We detail a comparison between AdaFM and VRAdaGDA Huang et al. (2023). Both algorithms employ similar estimators, but VRAdaGDA requires unique momentum parameters and learning rates for each variable. Specifically, VRAdaGDA sets $\beta_t^x = c_1(\eta_t^x)^2$ for $x$ and $\beta_t^y = c_2(\eta_t^y)^2$ for $y$, with $\eta_t^x = k\gamma/(m+t)^{1/3}$ and $\eta_t^y = k\lambda/(m+t)^{1/3}$. It is crucial to note that the settings of

$c_1, c_2, k, \gamma, \lambda$, and $m$ are all dependent on problem-dependent parameters, and the precise settings of these parameters are vital for the algorithm's convergence. This dependency significantly restricts the algorithm's practical application. We will further explore the algorithm's sensitivity to these parameter settings and the challenges of identifying the optimal parameter combination in experiments.

### 4.2 Analysis of the NC-PL Setting

The PL condition appears to relax the strongly convex or concave setting. Strongly Concave requires that the second derivative of the function (Hessian matrix) is negative definite over the entire domain, which is a strict assumption, while PL does not require the existence or nature of the second derivative. This kind of setting is often more common in machine learning Nouiehed et al. (2019); Huang et al. (2023); Huang (2023); Lei et al. (2017). Under the PL conditions, the variable $y$ may also be non-concave. Accordingly, we leverage the following assumption to indicate the PL condition and then present the corresponding convergence result.

**Assumption 5** (PL condition in $y$). *Assume function $f(x, y)$ satisfies $\mu_y$-PL condition in variable $y$ for any fixed $x \in \mathbb{R}^{d_1}$ and $y \in \mathcal{Y}$, such that*

$$\|\nabla_y f(x,y)\|^2 \geq 2\mu_y \left( \max_{y^*} f(x, y^*) - f(x, y) \right).$$

**Theorem 2** (Convergence of NC-PL). *Under Assumptions 1-3 and 5, after $T$ training epochs, AdaFM in Algorithm 1 satisfies*

$$\mathbb{E} \sum_{t=1}^{T} \|\nabla_x f(x_t, y_t)\|^2 + \mathbb{E} \sum_{t=1}^{T} \|\nabla_y f(x_t, y_t)\|^2 = O \left( \kappa^{\frac{20}{3(1-\delta_x)}} T^{\frac{2\delta_x}{3(1-\delta_x)}} + \kappa^{\frac{10}{3\delta_y}} T^{\frac{1-2\delta_y}{3\delta_y}} \right).$$

*Then according to the setting of $\delta_x$ and $\delta_y$, we can get*

$$\frac{1}{T} \left[ \mathbb{E} \sum_{t=1}^{T} \|\nabla_x f(x_t, y_t)\| + \mathbb{E} \sum_{t=1}^{T} \|\nabla_y f(x_t, y_t)\| \right] = O \left( \frac{\kappa^5}{T^{1/3+\delta}} \right).$$

In this setting, obtaining a direct upper bound for $\mathbb{E} \sum_{t=1}^{T} \|\nabla_y f(x_t, y_y)\|^2$ proves challenging due to the absence of the strong concavity condition. However, by leveraging the smoothness properties of both variables and the $\mu_y$-PL condition, we can establish an upper bound for $\mathbb{E} \sum_{t=1}^{T} [\Phi(x_t) - f(x_t, y_t)]$. Furthermore, we can transform this into $\mathbb{E} \sum_{t=1}^{T} [\|\nabla_x f(x_t, y_t)\|^2]$ using the quadratic growth condition Karimi et al. (2016), which is the condition is interchangeable with the $\mu_y$-PL condition. It allows us to derive the final result. Therefore, modifying this setting solely affects the upper bound of $\mathbb{E} \sum_{t=1}^{T} \|\nabla_y f(x_t, y_t)\|^2$.

**Remark 4.** In Theorem 2, AdaFM achieves a convergence rate close to $O(\kappa^5/T^{1/3+\delta})$, with the total number of training epochs required such that the iteration $T$ is arbitrarily close to $O(1/\epsilon^{-3})$ under the setting that $\delta$ is close to 0. Although the NC-PL setting is more strict than NC-SC, we can see that AdaFM's performance is only slightly below the rate of $O(\kappa^{4.5}/T^{1/3+\delta})$ in the NC-SC setting, demonstrating its effectiveness under the NC-PL setting. This highlights the scalability of AdaFM and affords many different machine learning scenarios. The slight performance drop occurs because we use the PL condition to deduce $\mathbb{E} \sum_{t=1}^{T} [\Phi(x_t) - f(x_t, y_t)]$ from $\mathbb{E} \sum_{t=1}^{T} [\|\nabla_x f(x_t, y_t)\|^2]$ rather than directly obtaining its upper bound from the strongly-concave condition. To the best of our knowledge, AdaFM is the first algorithm to achieve parameter-free optimization under the NC-PL setting while also nearing the optimal convergence rate Huang (2023).

## 5 Experiments

In this section, we evaluate the performance of our proposed AdaFM algorithm compared to RSGDA Huang & Gao (2023), VRAdaGDA Huang et al. (2023), and TiAda Li et al. (2023a) under three different learning tasks: (1) a test function with synthetic datasets, (2) optimizing the deep AUC loss (an NC-SC objective) in Yuan et al. (2021), and (3) training the NC-PL objective on Wasserstein-GAN with Gradient Penalty (WGAN-GP) Sinha et al. (2018). In this paper, we uniformly denote the initial

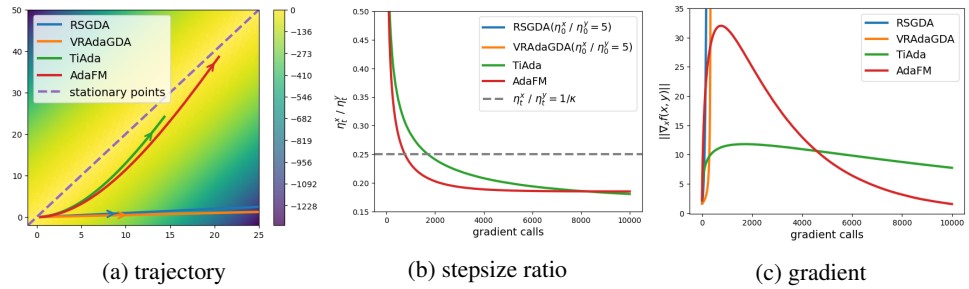

(a) trajectory  (b) stepsize ratio  (c) gradient

Figure 2: Numerical results on the test function $f(x,y) = \frac{1}{2}y^2 + Lxy - \frac{L^2}{2}x^2$, where $L = 2$.

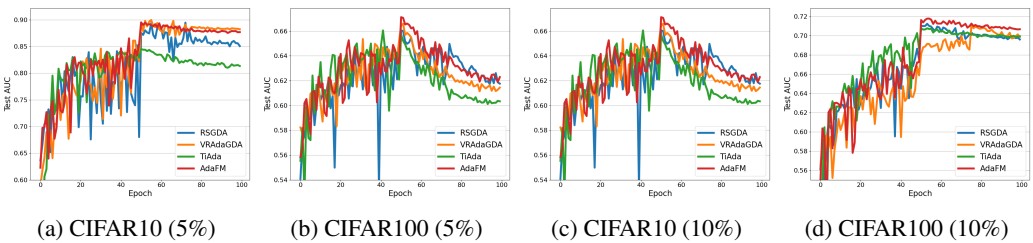

(a) CIFAR10 (5%)  (b) CIFAR100 (5%)  (c) CIFAR10 (10%)  (d) CIFAR100 (10%)

Figure 3: Convergence curves of deep AUC with an imbalance ratio of 5% and 10%.

learning rates for variables $x$ and $y$ as $\gamma$ and $\lambda$ respectively, for the aforementioned algorithms, to ensure clarity. It is worth noting that setting the initial learning rate does not imply that the learning rate will remain unchanged during the iteration. Additional experimental setups and results will be deferred to Appendix A in detail.

## 5.1 TEST FUNCTIONS

We use the example $f(x,y) = \frac{1}{2}y^2 + Lxy - \frac{L^2}{2}x^2$, proposed in TiAda Li et al. (2023a), to evaluate the four algorithms. We adopt the same setting as in TiAda, i.e., $\gamma/\lambda = 5$, $L = 2$, and introduce a small amount of noise into the gradient. We set $\delta = 0.1$ in all toy examples. We select the initial point as $(0.1, 0)$. As Figure 2a depicts, both TiAda and AdaFM manage this poor initial stepsize ratio effectively, while VRAdaGDA and RSGDA struggle to converge. Figure 2b illustrates that, both TiAda and AdaFM are able to adaptively adjust the stepsize to the desired ratio, i.e., $1/\kappa$, and it can be seen that AdaFM adjusts the stepsize ratio more quickly. In contrast, RSGDA and VRAdaGDA do not inherently have the ability to dynamically adjust the stepsize ratio. Moreover, as can be seen in Figure 2c, AdaFM approaches the stationary points more quickly after a relatively large initial divergence. However, TiAda approaches the stationary points at a very slow rate, even though it can adaptively adjust the learning rate. In addition, RSGDA and VRAdaGDA exhibit divergences.

## 5.2 DEEP AUC

An impactful application of the minimax problem is to optimize margin-based min-max surrogate losses, which can be considered as deep AUC maximization. In situations where imbalanced datasets can skew a model's performance metrics, the optimization of AUC scores has paramount significance. The the AUC margin Loss Yuan et al. (2021) is formulated as follows:

$$\min_{\mathbf{x}\in\mathbb{R}^{d_1}, (a,b)\in\mathbb{R}^2} \max_{y\in\mathcal{Y}} f(x,a,b,y) := \mathbb{E}_\xi[F(x,a,b,y;\xi)]. \tag{8}$$

The experimental results shown in Figure 3 were conducted on the CIFAR10 and CIFAR100 datasets with an imbalance ratio of 5% and 10%. It can be observed that under more challenging conditions, specifically when the imbalance ratio is 5%, TiAda performs very poorly on both CIFAR10 and CIFAR100. Compared to the best-performing algorithm, TiAda's AUC on the two datasets was 5% and 2% lower, respectively. Notably, RSGDA is highly unstable during the training process, experiencing severe drops in performance across all four scenarios. Although hyperparameter

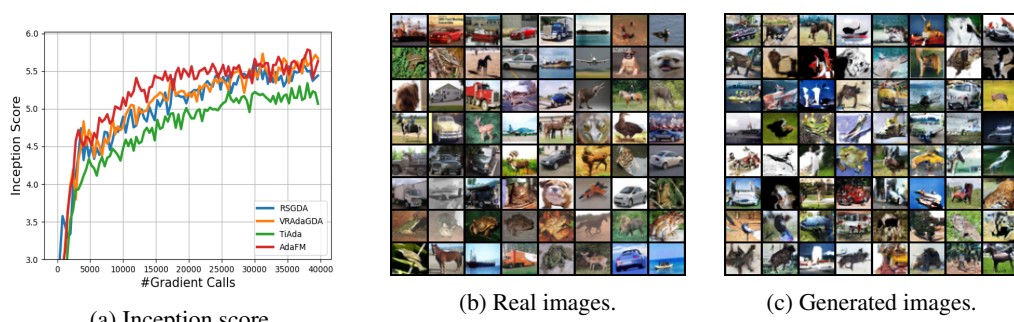

(a) Inception score.

(b) Real images.

(c) Generated images.

Figure 4: Inception score and visualization from WGAN-GP on CIFAR10.

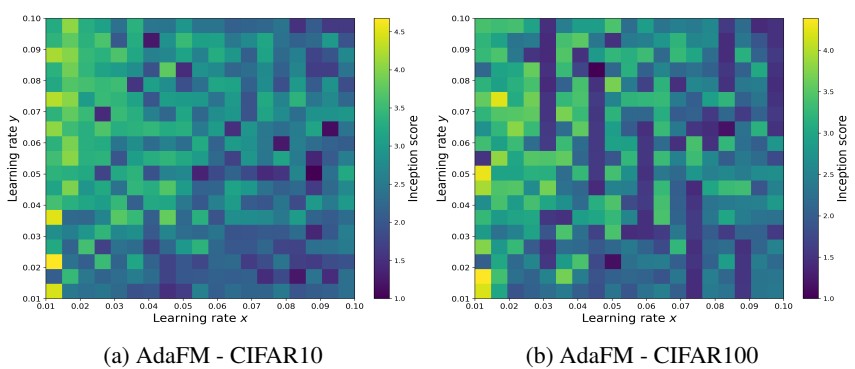

(a) AdaFM - CIFAR10

(b) AdaFM - CIFAR100

Figure 5: The hyperparameter grid search of AdaFM.

searches were conducted on the learning rates of all four algorithms using the same step size, and an additional hyperparameter search was performed for the momentum parameters in the case of RSGDA and VRAdaGDA, AdaFM consistently outperforms the others in almost cases.

### 5.3 WGAN-GP

Generative Adversarial Networks (GANs), as elucidated in Arjovsky et al. (2017), exemplify the efficacy of minimax optimization in the realm of machine learning. Conventionally, a discriminator network discerns whether an image originates from the authentic dataset, while a generator crafts images that are virtually indistinguishable from genuine dataset images, effectively 'deceiving' the discriminator. We employed the WGAN-GP loss proposed by Sinha et al. (2017) on the CIFAR10 dataset to enhance discriminator performance. Further findings on CIFAR100 utilizing the WGAN-GP approach are expounded in Appendix A, showcasing its efficacy across various datasets.

Figure 4a display inception scores on WGAN-GP. At the start of training, the inception score drops, likely due to updating the discriminator once per iteration, weakening its early discriminatory ability. However, as training continues, the discriminator improves, enhancing the generator's performance and leading to a rise in the inception score. Notably, AdaFM not only outperforms these algorithms but also achieves higher scores more rapidly and consistently as it converges. In contrast, TiAda's inception score is approximately 0.5 points lower than those of the other algorithms. Besides, Figures 4b-4c present a set of real samples from CIFAR10 alongside samples generated by AdaFM, showcasing its effectiveness in generating high-quality images.

In addition, we compared the hyperparameter grid search results of RSGDA and AdaFM within the same intervals. The hyperparameter grid search was performed in the range [0, 0.1] with a step size of 0.005, as shown in Figure 5. It can be observed that within this parameter space, AdaFM performs well for the vast majority of parameter combinations, while RSGDA struggles to train the model. Additionally, AdaFM's inception score significantly exceeds that of RSGDA.

## 6 CONCLUSION

In this paper, we present AdaFM, an adaptive variance-reduced algorithm that eliminates the need for manual hyper-parameter tuning, improving the practical application of variance-reduction techniques in stochastic minimax optimizations. AdaFM uniquely adjusts momentum parameters based on iteration count and adaptively modifies learning rates using historical estimator information combined with momentum parameters. Although the theoretical result in the NC-PL setting is $O(\kappa^5 T^{-1/3})$, which is worse than the NC-SC setting's $O(\kappa^{4.5} T^{-1/3})$ due to the more complex properties, both achieve an $\epsilon$-stationary point with an optimal complexity of $O(\epsilon^{-3})$, which align the best results among existing parametric algorithms. Extensive experimental evidence validates the effectiveness and robustness of AdaFM across various scenarios. In the future, we aim to develop parameter-free algorithms for more complex scenarios, e.g., minimax optimization without projection and compositional minimax optimizations, and relax conditions, e.g., non-convex non-concave settings.

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

## A  ADDITIONAL EXPERIMENTAL

### A.1  RESULTS OF ADDITIONAL TEST FUNCTIONS

In addition to the test functions presented in Sections 5, we have incorporated one additional test results to further validate the robustness and versatility of our AdaFM algorithm. To emulate stochastic gradient behavior, we introduced Gaussian noise with a mean of 0 and a variance of 0.1 to the function gradients of both the primal variable $x$ and the dual variable $y$. $r = \gamma/\lambda$ is the initial stepsize ratio. We chose the $r = 1/0.01$, $r = 1/0.03$ and $r = 1/0.05$ settings aligned with TiAda. It can be observed that AdaFM performs best across all three learning rate ratios, whereas TiAda only adapts its learning rate very slowly, approaching the optimal point at a sluggish pace. It is also worth noting that with less appropriate learning rate ratios, such as $r = 1/0.05$, RSGDA and VRAdaGDA exhibit worse performance at the beginning of the iteration due to their inability to adjust the learning rate ratios adaptively, as shown in Figure 6c.

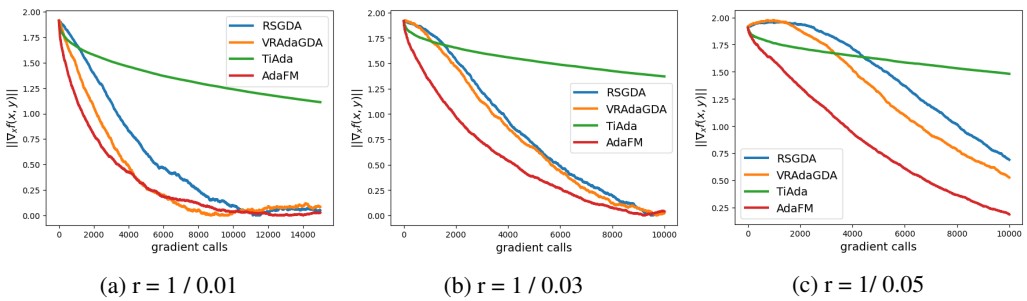

(a) r = 1 / 0.01     (b) r = 1 / 0.03     (c) r = 1/ 0.05

Figure 6: Results on McCormick function $f(x, y) = \sin(x + y) + (x - y)^2 - 1.5x + 2.5y + 1$.

### A.2  EXPERIMENTAL SETUPS

### A.2.1  SETUPS OF DEEP AUC

To generate imbalanced data, we utilized the approach described by Yuan et al. (2021). In particular, we divided the training data into two equal portions based on class ID, designating them as positive and negative classes. We then randomly eliminated certain samples from the positive class to create the imbalance, while the testing set remained unchanged. Our experiments were conducted using ResNet20, and we examined imbalance ratios of 5%, 10%, and 30%. For AdaFM, we set $\delta$ to 0.001. For TiAda, we set $\alpha$ and $\beta$ to $0.5 + 0.001$ and $0.5 - 0.001$. To further demonstrate AdaFM's ease of implementation, we limited the hyperparameter search to a narrow range for both TiAda and AdaFM.

702
703
704
705
706
707
708
709
710

Specifically, we searched for the initial learning rate $\gamma$ within $[0.1, 0.5]$ using a step size of 0.1, and for $\lambda$ within $[0.6, 1.0]$ with the same step size. For RSGDA and AdaFM, the search range for both $\gamma$ and $\lambda$ was $[0.1, 1]$ with a finer step size of 0.05. Additionally, we searched within $[0.05, 0.95]$ in increments of 0.05 for both their $\beta_x$ and $\beta_y$. The decay rate was applied at 50% and 75% of the total training duration, consistent with the settings in Yuan et al. (2022). The batch size was standardized at 128 for all datasets, and a weight decay of $1e$-4 was uniformly implemented across all methodologies.

### A.2.2 Setups of W-GAN

711
712
713
714
715
716
717
718
719
720
721
722
723
724
725
726

In this section, we adapted the code from Li et al. (2023a) for our experiments. For the implementation, we used a four-layer CNN for the discriminator and another four-layer CNN with transpose convolution layers for the generator, following the architecture specified in Daskalakis et al. (2018). We set the batch size to 512, the dimension of the latent variable to 50, and assigned a weight of $10^{-4}$ for the gradient penalty term. To compute the inception score, we utilized a pre-trained inception network, processing 8,000 synthesized samples. Since all the optimizers mentioned above are one-loop algorithms, we updated the discriminator only once for each generator to ensure a fair comparison. On CIFAR10 and CIFAR100, we performed 40,000 iterations on both the discriminators and generators. For AdaFM, we set $\delta$ to 0.001, while for TiAda, we set $\alpha$ and $\beta$ to $0.5 + 0.001$ and $0.5 - 0.001$, respectively. For several algorithms, we selected different hyperparameter search ranges. Specifically, we performed a hyperparameter search for RSGDA and VRAdaGDA's learning rates for both $x$ and $y$, using a step size of 0.0002 within the range of 0 to 0.01, while the hyperparameter search for $\beta_x$ and $\beta_y$ ranged from 0.5 to 0.9 in steps of 0.1. Figure 1 in section 1 shows the case of $\beta_x = \beta_y = 0.9$ after 10,000 iterations. Similarly, Figure 5 shows the inception score after 10,000 iteration, swith the hyperparameters search for $\gamma$ and $\lambda$ ranging from 0 to 0.1 in steps of 0.005 for AdaFM.

727
728
729
730

### A.3 Additional Results on Realistic Machine learning scenarios and Datasets

### A.3.1 Additional Deep AUC Results

731
732
733
734
735
736
737

We conducted another experiment on both CIFAR-10 and CIFAR-100 with a 30% imbalance ratio, as shown in Figure 7. It can be noticed from Figure 7a that both RSGDA and VRAdaGDA are very unstable during the training process, with large fluctuations in the training curves. In addition, due to the 30% imbalance ratio at this time, the task is relatively simple, and the four algorithms do not differ significantly in performance.

738
739
740
741
742
743
744
745
746

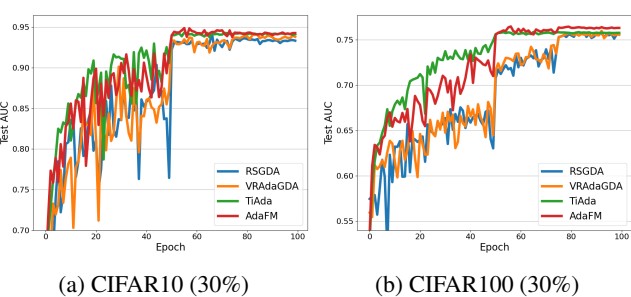

(a) CIFAR10 (30%)   (b) CIFAR100 (30%)

747
748
749
750

Figure 7: Convergence curves of deep AUC on CIFAR10 with an imbalance ratio of 30%.

### A.3.2 Additional WGAN-GP Results

751
752
753
754
755

We similarly tested the performance of the four algorithms on CIFAR100. It can be observed that AdaFM achieves the highest inception score in this case as well, while TiAda performs significantly worse than the other three algorithms, as shown in 8a. Figures 8b and 8c show a set of real images from CIFAR100 and a set of images generated by AdaFM training, respectively.

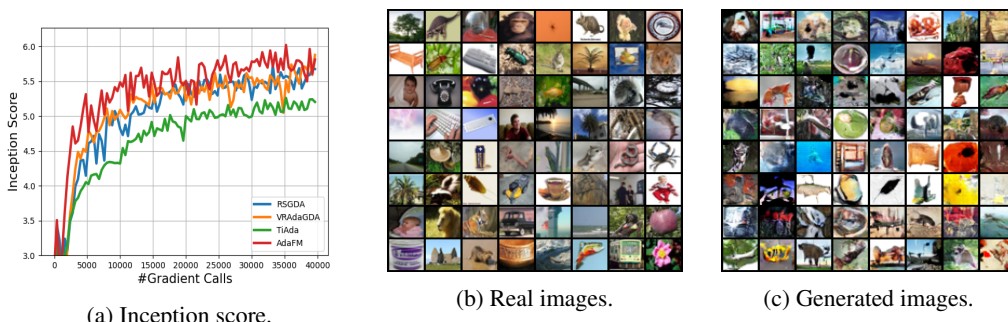

(a) Inception score.

(b) Real images.

(c) Generated images.

Figure 8: Inception score on CIFAR100.

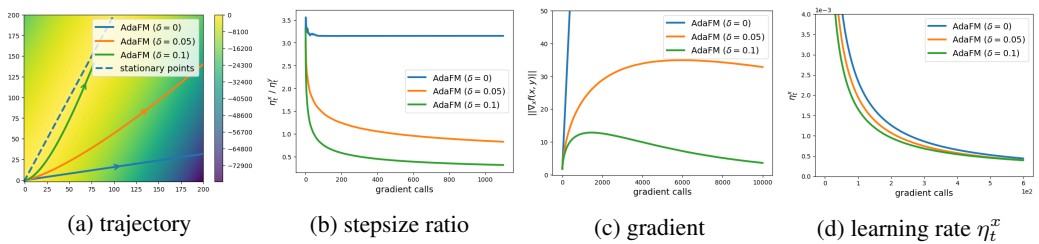

(a) trajectory

(b) stepsize ratio

(c) gradient

(d) learning rate $\eta_t^x$

Figure 9: Ablation Study on the test function

## A.4 ABLATION STUDY ON $\delta$

In this section, we demonstrate the effect of $\delta$ on the algorithm. From the settings of $\eta_t^x$ and $\eta_t^y$, it can be observed that an increase in the value of $\delta$ further reduces the learning rate of $x$ while increasing the learning rate of $y$. This adjustment causes the learning rates of $x$ and $y$ to reach the desired ratio more quickly in some scenarios. However, due to the rapid decrease in the learning rate of $x$, it may also slow down the overall convergence rate.

We use the same test function as shown in Figure 2, which helps us visualize the role of $\delta$. It can be observed that AdaFM fails to converge at $\delta = 0$, as shown in Figure 9a, and loses the ability to adaptively control the stepsize ratio, as shown in Figure 9b. As $\delta$ increases, AdaFM adjusts more effectively, and the trajectory curve approaches the stationary points with greater curvature. Meanwhile, the stepsize ratio reaches the desired value more quickly. However, this also causes the learning rate of $x$ to decrease more rapidly, as illustrated in Figure 9d.

In addition, we show the effect of $\delta$ under a complex task, i.e., training WGAN-GP. By simply choosing $\gamma = \lambda = 0.005$, and varying $\delta$ in the range of [0.1, 0.2, 0.3], as shown in Figure 10. We can find that in this case, the smaller the value of $\delta$, the better inception socre.

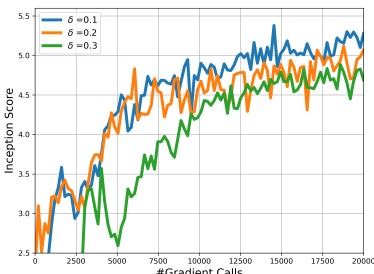

Figure 10: Ablation study on WGAN-GP

## B  USEFUL LEMMAS

**Lemma 1.** *(Lemma A.2 in Yang et al. (2022b)) Let $x_1, \cdots, x_T$ be a sequence of non-negative real numbers, $\alpha \in (0, 1)$, then we have:*

$$\left(\sum_{t=1}^{T} x_t\right)^{1-\alpha} \leq \sum_{t=1}^{T} \frac{x_t}{\left(\sum_{k=1}^{t} x_k\right)^{\alpha}} \leq \frac{1}{1-\alpha}\left(\sum_{t=1}^{T} x_t\right)^{1-\alpha}.$$

**Lemma 2.** *(Lemma A.5 in Nouiehed et al. (2019)) Under Assumptions 1 and 5, we have*

$$\|\nabla\Phi(x_1) - \nabla\Phi(x_2)\| \leq L_\Phi \|x_1 - x_2\|, \ \forall x_1, x_2$$

*where $L_\Phi = L + \frac{\kappa L}{2}$.*

## C  ANALYSIS OF THEOREM 1

In this section, we reiterate our primary goal of pinpointing a near-stationary point for the minimax problem, represented by $\mathbb{E}[\|\nabla_x f(x, y)\|] \leq \epsilon$ and $\mathbb{E}[\|\nabla_y f(x, y)\|] \leq \epsilon$. Here, the expectation incorporates every element of algorithmic randomness, ensuring a comprehensive and nuanced understanding of the system's behavior amidst varying conditions and inputs.

### C.1  INTERMEDIATE LEMMAS OF THEOREM 1

we first consider the detailed proof of the term $\epsilon_t^x$.

**Lemma 3.** *Under Assumptions 1-2, the error dynamic $\mathbb{E}[\sum_{t=1}^{T} \|\epsilon_t^x\|^2]$ can be upper-bounded as follows:*

$$\mathbb{E}\sum_{t=1}^{T} \|\epsilon_t^x\|^2 \leq 24G^2 T^{\frac{1}{3}} + \frac{24\gamma^2}{1-2\delta_x} T^{\frac{2-4\delta_x}{3}} (\mathbb{E}\sum_{t=1}^{T-1} \|v_t\|^2)^{1-2\delta_x} + \frac{24\lambda^2}{1-2\delta_y} T^{\frac{2-4\delta_y}{3}} (\mathbb{E}\sum_{t=1}^{T-1} \|w_t\|^2)^{1-2\delta_y}.$$

*Proof of Lemma 3.* According to equation 6, we can get

$$\epsilon_t^x = (1 - \beta_t)\epsilon_{t-1}^x + (1 - \beta_t)Z_t^x + \beta_t(\nabla_x f(x_t, y_t; \xi_t^x) - \nabla_x f(x_t, y_t)),$$

where $Z_t^x = (\nabla_x f(x_t, y_t; \xi_t^x) - \nabla_x f(x_{t-1}, y_{t-1}; \xi_t^x)) - (\nabla_x f(x_t, y_t) - \nabla_x f(x_{t-1}, y_{t-1}))$.

Taking the square of the above equation, we have:

$$\mathbb{E}\left[\|\epsilon_t^x\|^2\right]$$

$$\leq (1 - \beta_t)^2 \mathbb{E}\left[\|\epsilon_{t-1}^x\|^2\right] + \|(1 - \beta_t)Z_t^x + \beta_t(\nabla_x f(x_t, y_t; \xi_t^x) - \nabla_x f(x_t, y_t))\|^2$$

$$\leq (1 - \beta_t)^2 \mathbb{E}\left[\|\epsilon_{t-1}^x\|^2\right] + 2(1 - \beta_t)^2 \|Z_t^x\|^2 + 2\beta_t^2 \mathbb{E}\left[\|\nabla_x f(x_t, y_t; \xi_t^x) - \nabla_x f(x_t, y_t)\|^2\right]$$

$$\leq (1 - \beta_t)\mathbb{E}\left[\|\epsilon_{t-1}^x\|^2\right] + 8L^2 \mathbb{E}\left[(\eta_{t-1}^x)^2 \|v_{t-1}\|^2\right] + 8L^2 \mathbb{E}\left[(\eta_{t-1}^y)^2 \|w_{t-1}\|^2\right] + 4\beta_t^2 G^2.$$

Dividing above inequality by $\beta_t$, and re-arranging implies:

$$\mathbb{E}\sum_{t=1}^{T} \|\epsilon_{t-1}\|^2 \leq \underbrace{-\frac{\mathbb{E}[\|\epsilon_T\|^2]}{\beta_T}}_{\text{(i)}} + \underbrace{\sum_{t=1}^{T-1}\left(\frac{1}{\beta_{t+1}} - \frac{1}{\beta_t}\right)\mathbb{E}[\|\epsilon_t\|^2]}_{\text{(ii)}} + \underbrace{4G^2 \sum_{t=1}^{T} \beta_t}_{\text{(iii)}}$$

$$+ \underbrace{8L^2 \mathbb{E}[\sum_{t=1}^{T} \frac{(\eta_{t-1}^x)^2 \|v_{t-1}\|^2}{\beta_t}] + 8L^2 \mathbb{E}[\sum_{t=1}^{T} \frac{(\eta_{t-1}^y)^2 \|w_{t-1}\|^2}{\beta_t}]}_{\text{(iv)}}. \tag{9}$$

Then we bound the term on the RHS of above inequality.

**Bounding the term** (i). Since $\beta_T \leq 1$, we can get $-\frac{\mathbb{E}[\|\epsilon_T\|^2]}{\beta_T} \leq -\mathbb{E}[\|\epsilon_T\|^2]$.

**Bounding the term** (ii). Note that $g(a) = z^{2/3}$ is a concave function in $\mathbb{R}_+$. Thus we can get for any $a_1, a_2 \geq 0$, $(a_1 + a_2)^{2/3} - a_1^{2/3} \leq \frac{2}{3}a_1^{-1/3}a_2$. Therefore, for any $t \geq 2$, we can get

$$\frac{1}{\beta_{t+1}} - \frac{1}{\beta_t} = t^{2/3} - (t-1)^{2/3} \leq \frac{2}{3}(t-1)^{-1/3} \leq \frac{2}{3}.$$

Then we can get (ii) $\leq \frac{2}{3}\mathbb{E}[\|\epsilon_t\|^2]$.

**Bounding the term** (iii). According to the definition of $\beta_t$, we can get

$$\sum_{t=1}^{T} \beta_t = 1 + \sum_{t=1}^{T-1} \frac{1}{t^{2/3}} \leq 1 + 3T^{1/3} \leq 4T^{1/3},$$

where the first inequality holds by Lemma 3 in Levy et al. (2021), i.e., let $b_1, \cdots, b_n \in (0, b]$ be a sequence of non-negative real numbers for some positive real number $b, b_0 > 0$ and $p \in (0, 1]$ a rational number, then,

$$\sum_{i=1}^{n} \frac{b_i}{\left(b_0 + \sum_{j=1}^{i-1} b_j\right)^p} \leq \frac{b}{(b_0)^p} + \frac{2}{1-p}\left(b_0 + \sum_{i=1}^{n} b_i\right)^{1-p}.$$

**Bounding the term** (iv). According to the definition of $\eta_t^x$, we can get

$$\mathbb{E}\left[\sum_{t=1}^{T} \frac{(\eta_{t-1}^x)^2 \|v_{t-1}\|^2}{\beta_t}\right] = \gamma^2 \mathbb{E}\left[\sum_{t=1}^{T} \frac{\|v_{t-1}\|^2/\beta_t}{(\sum_{i=1}^{t-1}\|v_i\|^2/\beta_{i+1})^{2\delta_x}}\right] \leq \frac{\gamma^2}{1-2\delta_x}\mathbb{E}\left[(\sum_{t=1}^{T-1}\frac{\|v_t\|^2}{\beta_{t+1}})^{1-2\delta_x}\right]$$

$$\leq \frac{\gamma^2}{1-2\delta_x}T^{\frac{2-4\delta_x}{3}}(\mathbb{E}\sum_{t=1}^{T-1}\|v_t\|^2)^{1-2\delta_x}.$$

Similarly, we can get

$$\mathbb{E}\left[\sum_{t=1}^{T} \frac{(\eta_{t-1}^y)^2 \|w_{t-1}\|^2}{\beta_t}\right] = \lambda^2 \mathbb{E}\left[\sum_{t=1}^{T} \frac{\|w_{t-1}\|^2/\beta_t}{(\sum_{i=1}^{t-1}\|w_i\|^2/\beta_{i+1})^{2\delta_y}}\right] \leq \frac{\lambda^2}{1-2\delta_y}\mathbb{E}\left[(\sum_{t=1}^{T-1}\frac{\|w_t\|^2}{\beta_{t+1}})^{1-2\delta_y}\right]$$

$$\leq \frac{\lambda^2}{1-2\delta_y}T^{\frac{2-4\delta_y}{3}}(\mathbb{E}\sum_{t=1}^{T-1}\|w_t\|^2)^{1-2\delta_y}.$$

Plugging above bounds into equation 9, we can get

$$\mathbb{E}\sum_{t=1}^{T}\|\epsilon_t^x\|^2 \leq 48G^2T^{\frac{1}{3}} + \frac{24\gamma^2}{1-2\delta_x}T^{\frac{2-4\delta_x}{3}}(\mathbb{E}\sum_{t=1}^{T-1}\|v_t\|^2)^{1-2\delta_x} + \frac{24\lambda^2}{1-2\delta_y}T^{\frac{2-4\delta_y}{3}}(\mathbb{E}\sum_{t=1}^{T-1}\|w_t\|^2)^{1-2\delta_y}.$$

This complete the proof. $\qquad\square$

Since the error bounds in proving $\mathbb{E}\sum_{t=1}^{T}\|\epsilon_t^x\|^2$ and $\mathbb{E}\sum_{t=1}^{T}\|\epsilon_t^y\|^2$ are highly similar, we only need to give proof of one of them.

**Lemma 4.** *Under Assumptions 1-2, the error dynamic $\mathbb{E}[\sum_{t=1}^{T}\|\epsilon_t^y\|^2]$ can be upper-bounded as follows:*

$$\mathbb{E}\sum_{t=1}^{T}\|\epsilon_t^y\|^2 \leq 48G^2T^{\frac{1}{3}} + \frac{24\gamma^2}{1-2\delta_x}T^{\frac{2-4\delta_x}{3}}(\mathbb{E}\sum_{t=1}^{T-1}\|v_t\|^2)^{1-2\delta_x} + \frac{24\lambda^2}{1-2\delta_y}T^{\frac{2-4\delta_y}{3}}(\mathbb{E}\sum_{t=1}^{T-1}\|w_t\|^2)^{1-2\delta_y}.$$

Next we give the bound of $\sum_{t=1}^{T}\|\nabla_x f(x_t, y_t)\|^2$.

**Lemma 5.** *Under Assumptions 1-3 ,term $\sum_{t=1}^{T} \|\nabla_x f(x_t, y_t)\|^2$ can be upper-bounded as follows:*

$$\sum_{t=1}^{T} \|\nabla_x f(x_t, y_t)\|^2 \le \sum_{t=1}^{T} \|\epsilon_t^x\|^2 + 4\Phi_*(1/\beta_{T+1})^{\delta_x}(\sum_{t=1}^{T} \|v_t\|^2 + \|w_t\|^2)^{\delta_x} + \frac{L}{1-2\delta_x}(\sum_{t=1}^{T} \|v_t\|^2)^{1-2\delta_x}.$$

*Proof.* From Assumption 1 we know that $f(x, y)$ is smooth with respect to $x$, so we have:

$$f(x_{t+1}, y_t) - f(x_t, y_t) \le -\eta_t^x \langle \nabla_x f(x_t, y_t), v_t \rangle + \frac{L(\eta_t^x)^2}{2}\|v_t\|^2$$

$$\le -\eta_t^x \|\nabla_x f(x_t, y_t)\|^2 - \eta_t^x \langle \nabla_x f(x_t, y_t), \epsilon_t^x \rangle + \frac{L(\eta_t^x)^2}{2}\|v_t\|^2$$

$$\le -\frac{\eta_t^x}{2}\|\nabla_x f(x_t, y_t)\|^2 + \frac{\eta_t^x}{2}\|\epsilon_t^x\|^2 + \frac{L(\eta_t^x)^2}{2}\|v_t\|^2.$$

Define $\Delta_1 = f(x_1, y_1)$ and $\forall t \ge 2$,

$$\Delta_t = \left\{ \begin{array}{ll} f(x_t, y_{t-1}) + f(x_t, y_t), & f(x_t, y_t) \ge f(x_t, y_{t-1}), \\ f(x_t, y_t), & f(x_t, y_t) < f(x_t, y_{t-1}). \end{array} \right.$$

From Assumption 3 we can get $\Delta_t \le \|\Phi(x_t)\| + \|\Phi(x_{t-1})\| \le 2\Phi_*$. Re-arranging the above, and summing over $t$, we have:

$$\sum_{t=1}^{T} \|\nabla_x f(x_t, y_t)\|^2$$

$$\le \sum_{t=1}^{T} \frac{2}{\eta_t^x}(f(x_t, y_t) - f(x_{t+1}, y_t)) + \sum_{t=1}^{T} \|\epsilon_t^x\|^2 + \sum_{t=1}^{T} L\eta_t^x \|v_t\|^2$$

$$\le 2\sum_{t=2}^{T}(\frac{1}{\eta_t^x} - \frac{1}{\eta_{t-1}^x})\Delta_t - \frac{2\Delta_{T+1}}{\eta_T^x} + \sum_{t=1}^{T} \|\epsilon_t^x\|^2 + \sum_{t=1}^{T} L\eta_t^x \|v_t\|^2 \qquad (10)$$

$$\le \sum_{t=1}^{T} \|\epsilon_t^x\|^2 + \frac{4\Phi_*}{\eta_T^x} + L\sum_{t=1}^{T} \frac{\|v_t\|^2}{(\sum_{i=1}^{t} \|v_i\|^2)^{\delta_x}}$$

$$\le \sum_{t=1}^{T} \|\epsilon_t^x\|^2 + 4\Phi_*(1/\beta_{T+1})^{\delta_x}(\sum_{t=1}^{T} \|v_t\|^2 + \|w_t\|^2)^{\delta_x} + \frac{L}{1-\delta_x}(\sum_{t=1}^{T} \|v_t\|^2)^{1-\delta_x},$$

where the last second inequality holds by $\beta_t < 1$. This complete the proof. $\qquad \square$

Before bounding the term $\mathbb{E}\sum_{t=1}^{T} \|\nabla_y f(x_t, y_t)\|^2$, we first provide some useful lemmas.

**Lemma 6.** *Given Assumptions 1 to 4, if for $t = t_0$ to $t_1 - 1$ and any $\lambda_t > 0, S_t$,*

$$\|y_{t+1} - y_{t+1}^*\|^2 \le (1 + \lambda_t)\|y_{t+1} - y_t^*\|^2 + S_t,$$

*then we have:*

$$\mathbb{E}\left[\sum_{t=t_0}^{t_1-1}(f(x_t, y_t^*) - f(x_t, y_t))\right]$$

$$\le \mathbb{E}\left[\sum_{t=t_0+1}^{t_1-1}\left(\frac{2 - \eta_t^y \mu}{4\eta_t^y}\|y_t - y_t^*\|^2 - \frac{1}{2\eta_t^y(1+\lambda_t)}\|y_{t+1} - y_{t+1}^*\|^2\right)\right]$$

$$+ \mathbb{E}\left[\sum_{t=t_0}^{t_1-1}\frac{\eta_t^y}{2}\|w_t\|^2\right] + \mathbb{E}\left[\sum_{t=t_0}^{t_1-1}\frac{S_t}{2\eta_t^y(1+\lambda_t)}\right] + \mathbb{E}\left[\sum_{t=t_0}^{t_1-1}\frac{4}{\mu}\|\epsilon_t^y\|^2\right].$$

*Proof.* For any value of $\lambda_t > 0$, we have:

$$\|y_{t+1} - y_{t+1}^*\|^2$$
$$\leq (1 + \lambda_t)\|y_{t+1} - y_t^*\|^2 + S_t$$
$$= (1 + \lambda_t)\|\mathcal{P}_{\mathcal{Y}}(y_t + \eta_t^y w_t) - y_t^*\|^2 + S_t$$
$$\leq (1 + \lambda_t)\|y_t + \eta_t^y w_t - y_t^*\|^2 + S_t$$
$$\leq (1 + \lambda_t)\Big(\|y_t - y_t^*\|^2 + (\eta_t^y)^2\|w_t\|^2 + 2\eta_t^y\langle w_t, y_t - y_t^*\rangle + \eta_t^y\mu\|y_t - y_t^*\|^2 - \eta_t^y\mu\|y_t - y_t^*\|^2\Big) + S_t.$$

Rearranging the terms, we have:

$$\langle w_t, y_t^* - y_t\rangle - \frac{\mu}{2}\|y_t - y_t^*\|^2$$
$$\leq \frac{1 - \mu\eta_t^y}{2\eta_t^y}\|y_t - y_t^*\|^2 - \frac{1}{2\eta_t^y(1 + \lambda_t)}\|y_{t+1} - y_{t+1}^*\|^2 + \frac{\eta_t^y}{2}\|w_t\|^2 + \frac{S_t}{2\eta_t^y(1 + \lambda_t)}.$$

Then we can get

$$\langle \nabla_y f(x_t, y_t), y_t^* - y_t\rangle - \frac{\mu}{2}\|y_t - y_t^*\|^2$$
$$\leq \frac{1 - \mu\eta_t^y}{2\eta_t^y}\|y_t - y_t^*\|^2 - \frac{1}{2\eta_t^y(1 + \lambda_t)}\|y_{t+1} - y_{t+1}^*\|^2 + \frac{\eta_t^y}{2}\|w_t\|^2 + \frac{S_t}{2\eta_t^y(1 + \lambda_t)}$$
$$+ \langle \nabla_y f(x_t, y_t) - w_t, y_t^* - y_t\rangle$$
$$\leq \frac{1 - \mu\eta_t^y}{2\eta_t^y}\|y_t - y_t^*\|^2 - \frac{1}{2\eta_t^y(1 + \lambda_t)}\|y_{t+1} - y_{t+1}^*\|^2 + \frac{\eta_t^y}{2}\|w_t\|^2 + \frac{S_t}{2\eta_t^y(1 + \lambda_t)}$$
$$+ \frac{\mu}{4}\|y_t^* - y_t\|^2 + \frac{4}{\mu}\|\epsilon_t^y\|^2.$$

Using strongly concave we can get

$$\langle \nabla_y f(x_t, y_t), y_t^* - y_t\rangle - \frac{\mu}{2}\|y_t - y_t^*\|^2 \geq f(x_t, y_t^*) - f(x_t, y_t).$$

Telescoping from $t = t_0$ to $t - 1$, and taking the expectation we complete the proof. $\qquad\square$

**Lemma 7.** *Given Assumptions 1 to 2, we have:*

$$\mathbb{E}\left[\sum_{t=1}^{T}(f(x_t, y_t^*) - f(x_t, y_t))\right]$$
$$\leq \mathbb{E}\left[\sum_{t=2}^{T}\left(\frac{2 - \eta_t^y\mu}{4\eta_t^y}\|y_t - y_t^*\|^2 - \frac{1}{\eta_t^y(2 + \mu\eta_t^y)}\|y_{t+1} - y_{t+1}^*\|^2\right)\right] + \mathbb{E}\left[\sum_{t=1}^{T}\frac{4}{\mu}\|\epsilon_t^y\|^2\right]$$
$$+ \frac{\lambda}{2(1 - \delta_y)}\left(\mathbb{E}\sum_{t=1}^{T}\|w_t\|^2\right)^{1-\delta_y} + \frac{\kappa^2\gamma}{2\lambda(1 - \delta_x)G^{\delta_x - \delta_y}}\left(\mathbb{E}\sum_{t=1}^{T}\|v_t\|^2\right)^{1-\delta_x}$$
$$+ \frac{\kappa^2\gamma^2}{\lambda^2 G^{2\delta_x - 2\delta_y}}\left(\mathbb{E}\sum_{t=1}^{T}\|v_t\|^2\right).$$

*Proof.* By Young's inequality, we have:

$$\left\|y_{t+1} - y_{t+1}^*\right\|^2 \leq (1 + \lambda_t)\left\|y_{t+1} - y_t^*\right\|^2 + \left(1 + \frac{1}{\lambda_t}\right)\left\|y_{t+1}^* - y_t^*\right\|^2.$$

Then letting $\lambda_t = \frac{\mu \eta_t^y}{2}$ and by Lemma 6, we have:

$$\mathbb{E}\left[\sum_{t=1}^{T}\left(f\left(x_t, y_t^*\right) - f\left(x_t, y_t\right)\right)\right]$$

$$\leq \mathbb{E}\left[\sum_{t=2}^{T}\left(\frac{2 - \eta_t^y \mu}{4\eta_t^y}\left\|y_t - y_t^*\right\|^2 - \frac{1}{\eta_t^y\left(2 + \mu\eta_t^y\right)}\left\|y_{t+1} - y_{t+1}^*\right\|^2\right)\right]$$

$$+ \mathbb{E}\left[\sum_{t=1}^{T}\frac{\eta_t^y}{2}\left\|w_t\right\|^2\right] + \mathbb{E}\left[\sum_{t=1}^{T}\frac{4}{\mu}\|\epsilon_t^y\|^2\right] + \mathbb{E}\left[\sum_{t=1}^{T}\frac{(1 + \frac{2}{\mu\eta_t^y})}{\eta_t^y\left(2 + \mu\eta_t^y\right)}\|y_{t+1}^* - y_t^*\|^2\right].$$

We bound the term $\mathbb{E}\left[\sum_{t=1}^{T}\frac{(1 + \frac{2}{\mu\eta_t^y})}{\eta_t^y(2 + \mu\eta_t^y)}\|y_{t+1}^* - y_t^*\|^2\right]$.

$$\mathbb{E}\left[\sum_{t=1}^{T}\frac{(1 + \frac{2}{\mu\eta_t^y})}{\eta_t^y\left(2 + \mu\eta_t^y\right)}\|y_{t+1}^* - y_t^*\|^2\right] \leq \mathbb{E}\left[\sum_{t=1}^{T}\frac{(1 + \frac{2}{\mu\eta_t^y})}{2\eta_t^y}\|y_{t+1}^* - y_t^*\|^2\right]$$

$$\leq \kappa^2\mathbb{E}\left[\sum_{t=1}^{T}\frac{(1 + \frac{2}{\mu\eta_t^y})}{2\eta_t^y}(\eta_t^x)^2\|v_t\|^2\right] = \kappa^2\mathbb{E}\left[\sum_{t=1}^{T}\left(\frac{(\eta_t^x)^2}{2\eta_t^y} + \frac{(\eta_t^x)^2}{\mu(\eta_t^y)^2}\right)\|v_t\|^2\right]$$

$$= \kappa^2\mathbb{E}\left[\sum_{t=1}^{T}\left(\frac{\gamma}{2\lambda(\alpha_t^y)^{\delta_x - \delta_y}}\eta_t^x + \frac{\gamma^2}{\lambda^2(\alpha_t^y)^{2\delta_x - 2\delta_y}}\right)\|v_t\|^2\right]$$

$$\leq \frac{\kappa^2\gamma}{2\lambda(1 - \delta_x)(\alpha_1^y)^{\delta_x - \delta_y}}\left(\mathbb{E}\sum_{t=1}^{T}\|v_t\|^2\right)^{1-\delta_x} + \frac{\kappa^2\gamma^2}{\lambda^2(\alpha_1^y)^{2\delta_x - 2\delta_y}}\left(\mathbb{E}\sum_{t=1}^{T}\|v_t\|^2\right)$$

$$\leq \frac{\kappa^2\gamma}{2\lambda(1 - \delta_x)(\|w_1\|^2)^{\delta_x - \delta_y}}\left(\mathbb{E}\sum_{t=1}^{T}\|v_t\|^2\right)^{1-\delta_x} + \frac{\kappa^2\gamma^2}{\lambda^2(\|w_1\|^2)^{2\delta_x - 2\delta_y}}\left(\mathbb{E}\sum_{t=1}^{T}\|v_t\|^2\right)$$

$$= \frac{\kappa^2\gamma}{2\lambda(1 - \delta_x)G^{2(\delta_x - \delta_y)}}\left(\mathbb{E}\sum_{t=1}^{T}\|v_t\|^2\right)^{1-\delta_x} + \frac{\kappa^2\gamma^2}{\lambda^2 G^{4(\delta_x - \delta_y)}}\left(\mathbb{E}\sum_{t=1}^{T}\|v_t\|^2\right),$$

Combining the above two inequalities, we complete the proof. $\qquad\square$

**Lemma 8.** *Given Assumptions 1 to 2, we have*

$$\sum_{t=1}^{T}\left(\frac{2 - \eta_t^y \mu}{4\eta_t^y}\left\|y_t - y_t^*\right\|^2 - \frac{1}{\eta_t^y\left(2 + \mu\eta_t^y\right)}\left\|y_{t+1} - y_{t+1}^*\right\|^2\right)$$

$$\leq \left(\frac{G^{\frac{2}{3}}}{2\lambda} - \frac{\mu}{2}\right)\|y_0 - y_0^*\|^2 + \frac{G^2}{\mu^2\eta_T^y}.$$

*Proof.*

$$\sum_{t=1}^{T}\left(\frac{2 - \eta_t^y \mu}{4\eta_t^y}\left\|y_t - y_t^*\right\|^2 - \frac{1}{\eta_t^y\left(2 + \mu\eta_t^y\right)}\left\|y_{t+1} - y_{t+1}^*\right\|^2\right)$$

$$\leq \left(\frac{G^{\frac{2}{3}}}{2\lambda} - \frac{\mu}{2}\right)\|y_0 - y_0^*\|^2 + \frac{1}{2}\sum_{t=2}^{T-1}\left(\frac{1}{\eta_{t+1}^y} - \frac{1}{\eta_t^y}\right)\|y_t - y_t^*\|^2$$

$$\leq \left(\frac{G^{\frac{2}{3}}}{2\lambda} - \frac{\mu}{2}\right)\|y_0 - y_0^*\|^2 + \frac{1}{2\mu^2}\sum_{t=2}^{T-1}\left(\frac{1}{\eta_{t+1}^y} - \frac{1}{\eta_t^y}\right)\|\nabla_y f(x_t, y_t)\|^2$$

$$\leq \left(\frac{G^{\frac{2}{3}}}{2\lambda} - \frac{\mu}{2}\right)\|y_0 - y_0^*\|^2 + \frac{G^2}{2\mu^2}\sum_{t=2}^{T-1}\left(\frac{1}{\eta_{t+1}^y} - \frac{1}{\eta_t^y}\right)$$

$$\leq \left(\frac{G^{\frac{2}{3}}}{2\lambda} - \frac{\mu}{2}\right)\|y_0 - y_0^*\|^2 + \frac{G^2}{2\mu^2\eta_T^y},$$

where the second inequality holds by Assumption 4. This completes the proof. $\qquad\square$

**Lemma 9.** *Based on Lemmas 7 and 8, we can upper-bound* $\mathbb{E}\left[\sum_{t=1}^{T}\|\nabla_y f(x_t, y_t)\|^2\right]$ *as follows:*

$$
\mathbb{E}\left[\sum_{t=1}^{T}\|\nabla_y f(x_t, y_t)\|^2\right]
$$

$$
\leq \left(\frac{L\kappa G^{\frac{2}{3}}}{\lambda} - \mu L\kappa\right)\|y_0 - y_0^*\|^2 + 8\kappa^2 \mathbb{E}\left[\sum_{t=1}^{T}\|\epsilon_t^y\|^2\right] + \frac{\lambda L\kappa}{1-\delta_y}\mathbb{E}\left(\sum_{t=1}^{T}\|w_t\|^2\right)^{1-\delta_y}
$$

$$
+ \frac{\kappa^3 L\gamma}{\lambda(1-\delta_x)G^{2(\delta_x-\delta_y)}}\left(\mathbb{E}\sum_{t=1}^{T}\|v_t\|^2\right)^{1-\delta_x} + \frac{\kappa^3 L\gamma^2}{\lambda^2 G^{4(\delta_x-\delta_y)}}\left(\mathbb{E}\sum_{t=1}^{T}\|v_t\|^2\right)
$$

$$
+ \frac{\kappa^2 G^2}{\lambda^2 \mu}T^{2\delta_y/3}\left(\mathbb{E}\sum_{t=1}^{T}\|w_t\|^2\right)^{\delta_y}.
$$

*Proof.* Combining Lemma 7 and 8 we have:

$$
\mathbb{E}\left[\sum_{t=1}^{T}\left(f(x_t, y_t^*) - f(x_t, y_t)\right)\right]
$$

$$
\leq \left(\frac{G^{\frac{2}{3}}}{2\lambda} - \frac{\mu}{2}\right)\|y_0 - y_0^*\|^2 + \mathbb{E}\left[\frac{G^2}{2\mu^2\eta_T^y}\right] + \mathbb{E}\left[\sum_{t=1}^{T}\frac{4}{\mu}\|\epsilon_t^y\|^2\right]
$$

$$
+ \frac{\lambda}{2(1-\delta_y)}\left(\mathbb{E}\sum_{t=1}^{T}\|w_t\|^2\right)^{1-\delta_y} + \frac{\kappa^2\gamma}{2\lambda(1-\delta_x)G^{2(\delta_x-\delta_y)}}\left(\mathbb{E}\sum_{t=1}^{T}\|v_t\|^2\right)^{1-\delta_x}
$$

$$
+ \frac{\kappa^2\gamma^2}{\lambda^2 G^{4(\delta_x-\delta_y)}}\left(\mathbb{E}\sum_{t=1}^{T}\|v_t\|^2\right).
$$

According to the $\mu$ strongly concave in Assumption 4, we have:

$$
\mathbb{E}\left[\sum_{t=1}^{T}\|\nabla_y f(x_t, y_t)\|^2\right] \leq L^2\mathbb{E}\left[\sum_{t=1}^{T}\|y_t - y_t^*\|^2\right] \leq 2L\kappa\mathbb{E}\left[\sum_{t=1}^{T}\left(f(x_t, y_t^*) - f(x_t, y_t)\right)\right]
$$

Then we have:

$$
\mathbb{E}\left[\sum_{t=1}^{T}\|\nabla_y f(x_t, y_t)\|^2\right]
$$

$$
\leq \left(\frac{L\kappa G^{\frac{2}{3}}}{\lambda} - \mu L\kappa\right)\|y_0 - y_0^*\|^2 + 8\kappa^2 \mathbb{E}\left[\sum_{t=1}^{T}\|\epsilon_t^y\|^2\right] + \frac{\lambda L\kappa}{1-\delta_y}\left(\mathbb{E}\sum_{t=1}^{T}\|w_t\|^2\right)^{1-\delta_y}
$$

$$
+ \frac{\kappa^3 L\gamma}{\lambda(1-\delta_x)G^{2(\delta_x-\delta_y)}}\left(\mathbb{E}\sum_{t=1}^{T}\|v_t\|^2\right)^{1-\delta_x} + \frac{\kappa^3 L\gamma^2}{\lambda^2 G^{4(\delta_x-\delta_y)}}\left(\mathbb{E}\sum_{t=1}^{T}\|v_t\|^2\right) \tag{11}
$$

$$
+ \frac{\kappa^2 G^2}{\lambda^2 \mu}T^{2\delta_y/3}\left(\mathbb{E}\sum_{t=1}^{T}\|w_t\|^2\right)^{\delta_y}.
$$

This completes the proof. $\qquad\square$

## C.2 PROOF OF THEOREM 1

Now, we come to the proof of Theorem 1.

*Proof.* Due to the definition, we have $\|v_t\|^2 \leq 2\|\nabla_x f(x_t, y_t)\|^2 + 2\|\epsilon_t^x\|^2$ and $\|w_t\|^2 \leq 2\|\nabla_y f(x_t, y_t)\|^2 + 2\|\epsilon_t^y\|^2$. We divide the final part of the proof into four subcases. Introduce a constant $S$ and we will give the detailed definition later.

**Case 1:** Assume $\mathbb{E}\sum_{t=1}^T \|\nabla_x f(x_t, y_t)\|^2 \leq S\mathbb{E}\sum_{t=1}^T \|\epsilon_t^x\|^2$ and $\mathbb{E}\sum_{t=1}^T \|\nabla_y f(x_t, y_t)\|^2 \leq S\mathbb{E}\sum_{t=1}^T \|\epsilon_t^y\|^2$. Using the condition of this subcase implies

$$\mathbb{E}\sum_{t=1}^T (\|v_t\|^2 + w_t\|^2) \leq (2 + 2S)\mathbb{E}\sum_{t=1}^T (\|\epsilon_t^x\|^2 + \|\epsilon_t^y\|^2).$$

According to Lemma 3 and 4 we have:

$$\mathbb{E}\sum_{t=1}^T (\|\epsilon_t^x\|^2 + \|\epsilon_t^y\|^2) \leq 96G^2 T^{\frac{1}{3}} + \underbrace{\frac{48\gamma^2}{1 - 2\delta_x} T^{\frac{2-4\delta_x}{3}} (\mathbb{E}\sum_{t=1}^{T-1} \|v_t\|^2)^{1-2\delta_x}}_{(I)}$$

$$+ \underbrace{\frac{48\lambda^2}{1 - 2\delta_y} T^{\frac{2-4\delta_y}{3}} (\mathbb{E}\sum_{t=1}^{T-1} \|w_t\|^2)^{1-2\delta_y}}_{(II)} \quad (12)$$

According to Young's inequality, for any $a, b > 0$, and $p, q > 1 : \frac{1}{p} + \frac{1}{q} = 1$ we have $ab \leq \frac{a^p}{p} + \frac{b^q}{q}$. Setting $p = \frac{1}{2\delta_x}, q = \frac{1}{1-2\delta_x}$, we have

$$a^{\frac{2-4\delta_x}{3}} b^{1-2\delta_x} = \left(a\rho^{\frac{3}{2-4\delta_x}}\right)^{\frac{2-4\delta_x}{3}} \left(\frac{b}{\rho^{\frac{1}{1-2\delta_x}}}\right)^{1-2\delta_x}$$

$$\leq \frac{\left(a\rho^{\frac{3}{2-4\delta_x}}\right)^{\frac{(2-4\delta_x)p}{3}}}{p} + \frac{\left(\frac{b}{\rho^{\frac{1}{1-2\delta_x}}}\right)^{(1-2\delta_x)q}}{q} \quad (13)$$

$$= 2\delta_x a^{\frac{1-2\delta_x}{3\delta_x}} \rho^{\frac{1}{2\delta_x}} + \frac{(1 - 2\delta_x)b}{\rho^{\frac{1}{1-2\delta_x}}}.$$

It is also important to observe that the aforementioned inequality remains valid when substituting $\delta_x$ with $\delta_y$, i.e.,

$$a^{\frac{2-4\delta_y}{3}} b^{1-2\delta_y} \leq 2\delta_y a^{\frac{1-2\delta_y}{3\delta_y}} \rho^{\frac{1}{2\delta_y}} + \frac{(1 - 2\delta_y)b}{\rho^{\frac{1}{1-2\delta_y}}}.$$

Setting $\rho = (96\gamma^2(2 + 2S))^{1-2\delta_x}$ for Term (I) and $\rho = (96\lambda^2(2 + 2S))^{1-2\delta_y}$ for Term (II) we have:

$$\mathbb{E}\sum_{t=1}^T (\|\epsilon_t^x\|^2 + \|\epsilon_t^y\|^2)$$

$$\leq 96G^2 T^{\frac{1}{3}} + \frac{1}{2(2 + 2S)}\mathbb{E}\sum_{t=1}^T \|v_t\|^2 + \frac{1}{2(2 + 2S)}\mathbb{E}\sum_{t=1}^T \|w_t\|^2$$

$$+ \frac{96\gamma^2\delta_x}{1 - 2\delta_x}(96\gamma^2(2 + 2S))^{\frac{1-2\delta_x}{2\delta_x}} T^{\frac{1-2\delta_x}{3\delta_x}} + \frac{96\lambda^2\delta_y}{1 - 2\delta_y}(96\lambda^2(2 + 2S))^{\frac{1-2\delta_y}{2\delta_y}} T^{\frac{1-2\delta_y}{3\delta_y}}.$$

Denote $C_1 = \max\{\frac{96\gamma^2\delta_x}{1-2\delta_x}(96\gamma^2(2 + 2S))^{\frac{1-2\delta_x}{2\delta_x}}, \frac{96\lambda^2\delta_y}{1-2\delta_y}(96\lambda^2(2 + 2S))^{\frac{1-2\delta_y}{2\delta_y}}\}$, according to $1/2 > \delta_x > \delta_y > 0$, we have

$$\mathbb{E}\sum_{t=1}^T (\|\epsilon_t^x\|^2 + \|\epsilon_t^y\|^2)$$

$$\leq 96G^2 T^{\frac{1}{3}} + \frac{1}{2(2 + 2S)}\mathbb{E}\sum_{t=1}^T \|v_t\|^2 + \frac{1}{2(2 + 2S)}\mathbb{E}\sum_{t=1}^T \|w_t\|^2 + 2C_1 T^{\frac{1-2\delta_y}{3\delta_y}}.$$

Then we can get:

$$\frac{1}{2}\mathbb{E}\sum_{t=1}^{T}(\|\epsilon_t^x\|^2 + \|\epsilon_t^y\|^2) \leq 96G^2 T^{\frac{1}{3}} + 2C_1 T^{\frac{1-2\delta_y}{3\delta_y}}.$$

Above implies,

$$\mathbb{E}\sum_{t=1}^{T}\|\nabla_x f(x_t, y_t)\|^2 + \mathbb{E}\sum_{t=1}^{T}\|\nabla_y f(x_t, y_t)\|^2$$

$$\leq 2S\mathbb{E}\sum_{t=1}^{T}(\|\epsilon_t^x\|^2 + \|\epsilon_t^y\|^2) = O\Big(G^2 ST^{\frac{1}{3}} + C_1 ST^{\frac{1-2\delta_y}{3\delta_y}}\Big).$$

Moreover, according to $1/2 > \delta_x > \delta_y > 0$, we have $C_1 = O(S^{\frac{1-2\delta_y}{2\delta_y}})$. Then we can get

$$\mathbb{E}\sum_{t=1}^{T}\|\nabla_x f(x_t, y_t)\|^2 + \mathbb{E}\sum_{t=1}^{T}\|\nabla_y f(x_t, y_t)\|^2 = O(G^2 ST^{\frac{1}{3}} + S^{\frac{1}{2\delta_y}}T^{\frac{1-2\delta_y}{3\delta_y}}).$$

This complete the proof.

**Case 2:** Assume $\mathbb{E}\sum_{t=1}^{T}\|\nabla_x f(x_t, y_t)\|^2 \leq S\mathbb{E}\sum_{t=1}^{T}\|\epsilon_t^x\|^2$ and $\mathbb{E}\sum_{t=1}^{T}\|\nabla_y f(x_t, y_t)\|^2 \geq S\mathbb{E}\sum_{t=1}^{T}\|\epsilon_t^y\|^2$. Using the condition of this subcase implies

$$\mathbb{E}\sum_{t=1}^{T}\|v_t\|^2 \leq (2 + 2S)\mathbb{E}\sum_{t=1}^{T}\|\epsilon_t^x\|^2, \quad \mathbb{E}\sum_{t=1}^{T}\|w_t\|^2 \leq (2 + \frac{2}{S})\mathbb{E}\sum_{t=1}^{T}\|\nabla_y f(x_t, y_t)\|^2.$$

Combining Lemma 3 and 9, setting $C_2 = \min\{\frac{\lambda^2 G^{4(\delta_x - \delta_y)}}{16\kappa^3 L\gamma^2(2+2S)}, 1\}$ we have:

$$\mathbb{E}\sum_{t=1}^{T}\|\epsilon_t^x\|^2 + C_2\mathbb{E}\sum_{t=1}^{T}\|\nabla_y f(x_t, y_t)\|^2$$

$$\leq C_2\left(\frac{G^{\frac{2}{3}}}{2\lambda} - \frac{\mu}{2}\right)\|y_0 - y_0^*\|^2 + 8\kappa^2 C_2\mathbb{E}\left[\sum_{t=1}^{T}\|\epsilon_t^y\|^2\right] + \frac{C_2\lambda L\kappa}{1 - \delta_y}\left(\mathbb{E}\sum_{t=1}^{T}\|w_t\|^2\right)^{1-\delta_y}$$

$$+ \frac{C_2\kappa^3 L\gamma}{\lambda(1 - \delta_x)G^{2(\delta_x - \delta_y)}}\left(\mathbb{E}\sum_{t=1}^{T}\|v_t\|^2\right)^{1-\delta_x} + \frac{C_2\kappa^3 L\gamma^2}{\lambda^2 G^{4(\delta_x - \delta_y)}}\left(\mathbb{E}\sum_{t=1}^{T}\|v_t\|^2\right)$$

$$+ \frac{\kappa^2 G^2 C_2}{\lambda^2\mu}T^{2\delta_y/3}\left(\mathbb{E}\sum_{t=1}^{T}\|w_t\|^2\right)^{\delta_y} + 24G^2 T^{\frac{1}{3}} + \frac{24\gamma^2}{1 - 2\delta_x}T^{\frac{2-4\delta_x}{3}}(\mathbb{E}\sum_{t=1}^{T-1}\|v_t\|^2)^{1-2\delta_x}$$

$$+ \frac{24\lambda^2}{1 - 2\delta_y}T^{\frac{2-4\delta_y}{3}}(\mathbb{E}\sum_{t=1}^{T-1}\|w_t\|^2)^{1-2\delta_y}.$$

Using Case 2, we can get

$$
\mathbb{E}\sum_{t=1}^{T}\|\epsilon_t^x\|^2 + C_2\mathbb{E}\sum_{t=1}^{T}\|\nabla_y f(x_t, y_t)\|^2
$$

$$
\leq C_2\left(\frac{G^{\frac{2}{3}}}{2\lambda} - \frac{\mu}{2}\right)\|y_0 - y_0^*\|^2 + \frac{8\kappa^2 C_2}{S}\mathbb{E}\sum_{t=1}^{T}\|\nabla_y f(x_t, y_t)\|^2 + \frac{1}{16}\mathbb{E}\sum_{t=1}^{T}\|\epsilon_t^x\|^2
$$

$$
+ \frac{C_2\lambda L\kappa}{1-\delta_y}\left(\mathbb{E}\sum_{t=1}^{T}\|w_t\|^2\right)^{1-\delta_y} + \frac{C_2\kappa^3 L\gamma}{\lambda(1-\delta_x)G^{2(\delta_x-\delta_y)}}\left(\mathbb{E}\sum_{t=1}^{T}\|v_t\|^2\right)^{1-\delta_x}
$$

$$
+ \underbrace{\frac{\kappa^2 G^2 C_2}{\lambda^2\mu}T^{2\delta_y/3}\left(\mathbb{E}\sum_{t=1}^{T}\|w_t\|^2\right)^{\delta_y}}_{(III)} + \underbrace{\frac{24\gamma^2}{1-2\delta_x}T^{\frac{2-4\delta_x}{3}}(\mathbb{E}\sum_{t=1}^{T-1}\|v_t\|^2)^{1-2\delta_x}}_{(IV)}
$$

$$
+ \underbrace{\frac{24\lambda^2}{1-2\delta_y}T^{\frac{2-4\delta_y}{3}}(\mathbb{E}\sum_{t=1}^{T-1}\|w_t\|^2)^{1-2\delta_y}}_{(V)} + 24G^2 T^{\frac{1}{3}}.
$$

Setting $S \geq 16\kappa^2$, then we can get

$$
\frac{15}{16}\mathbb{E}\sum_{t=1}^{T}\|\epsilon_t^x\|^2 + \frac{C_2}{2}\mathbb{E}\sum_{t=1}^{T}\|\nabla_y f(x_t, y_t)\|^2
$$

$$
\leq C_2\left(\frac{G^{\frac{2}{3}}}{2\lambda} - \frac{\mu}{2}\right)\|y_0 - y_0^*\|^2 + \frac{C_2\lambda L\kappa}{1-\delta_y}\left(\mathbb{E}\sum_{t=1}^{T}\|w_t\|^2\right)^{1-\delta_y}
$$

$$
+ \frac{C_2\kappa^3 L\gamma}{\lambda(1-\delta_x)G^{2(\delta_x-\delta_y)}}\left(\mathbb{E}\sum_{t=1}^{T}\|v_t\|^2\right)^{1-\delta_x} + \underbrace{\frac{\kappa^2 G^2 C_2}{\lambda^2\mu}T^{2\delta_y/3}\left(\mathbb{E}\sum_{t=1}^{T}\|w_t\|^2\right)^{\delta_y}}_{(III)} \tag{14}
$$

$$
+ \underbrace{\frac{24\gamma^2}{1-2\delta_x}T^{\frac{2-4\delta_x}{3}}(\mathbb{E}\sum_{t=1}^{T-1}\|v_t\|^2)^{1-2\delta_x}}_{(IV)} + \underbrace{\frac{24\lambda^2}{1-2\delta_y}T^{\frac{2-4\delta_y}{3}}(\mathbb{E}\sum_{t=1}^{T-1}\|w_t\|^2)^{1-2\delta_y}}_{(V)} + 24G^2 T^{\frac{1}{3}}.
$$

According to Young's inequality, for any $a, b > 0$, and $p, q > 1 : \frac{1}{p} + \frac{1}{q} = 1$ we have $ab \leq \frac{a^p}{p} + \frac{b^q}{q}$.
Setting $p = \frac{1}{1-\delta_x}, q = \frac{1}{\delta_x}$, we have

$$
a^{\frac{2\delta_x}{3}}b^{\delta_x} = \left(a\rho^{\frac{3}{2\delta_x}}\right)^{\frac{2\delta_x}{3}}\left(\frac{b}{\rho^{\frac{1}{\delta_x}}}\right)^{\delta_x}
$$

$$
\leq \frac{\left(a\rho^{\frac{3}{2\delta_x}}\right)^{\frac{2\delta_x p}{3}}}{p} + \frac{\left(\frac{b}{\rho^{\frac{1}{\delta_x}}}\right)^{\delta_x q}}{q} \tag{15}
$$

$$
= (1-\delta_x)a^{\frac{2\delta_x}{3(1-\delta_x)}}\rho^{\frac{1}{1-\delta_x}} + \frac{\delta_x b}{\rho^{\frac{1}{\delta_x}}}.
$$

According to equation 15, setting $\rho = (\frac{8\kappa^2 G^2\delta_y(2+\frac{2}{S})}{\lambda^2\mu})^{\delta_y}$ for Term (III), we have:

$$
III \leq \frac{(1-\delta_y)\kappa^2 G^2 C_2}{\lambda^2\mu}(\frac{8\kappa^2 G^2\delta_x(2+\frac{2}{S})}{\lambda^2\mu})^{\frac{\delta_y}{1-\delta_y}}T^{\frac{2\delta_y}{3(1-\delta_y)}} + \frac{C_2}{8(2+\frac{2}{S})}\mathbb{E}\sum_{t=1}^{T}\|w_t\|^2. \tag{16}
$$

According to equation 13, setting $\rho = (288\gamma^2(2+2S))^{1-2\delta_x}$ for Term (IV) we can get

$$\text{IV} \le \frac{24\gamma^2}{1-2\delta_x}(288\gamma^2(2+2S))^{\frac{1-2\delta_x}{2\delta_x}}T^{\frac{1-2\delta_x}{3\delta_x}} + \frac{1}{8(2+2S)}\mathbb{E}\sum_{t=1}^{T}\|v_t\|^2. \tag{17}$$

According to equation 13, setting $\rho = (\frac{192\lambda^2(2+\frac{2}{S})}{C_2})^{1-2\delta_y}$ for Term (V) we can get

$$\text{V} \le \frac{24\lambda^2}{1-2\delta_y}(\frac{192\lambda^2(2+\frac{2}{S})}{C_2})^{\frac{1-2\delta_y}{2\delta_y}}T^{\frac{1-2\delta_y}{3\delta_y}} + \frac{C_2}{8(2+\frac{2}{S})}\mathbb{E}\sum_{t=1}^{T}\|w_t\|^2. \tag{18}$$

Then plugging equation 16 - equation 18 into equation 14, we can get

$$\frac{13}{16}\mathbb{E}\sum_{t=1}^{T}\|\epsilon_t^x\|^2 + \frac{C_2}{4}\mathbb{E}\sum_{t=1}^{T}\|\nabla_y f(x_t,y_t)\|^2$$

$$\le C_2\left(\frac{G^{\frac{2}{3}}}{2\lambda} - \frac{\mu}{2}\right)\|y_0 - y_0^*\|^2 + \frac{C_2\lambda L\kappa}{1-\delta_y}\left(\mathbb{E}\sum_{t=1}^{T}\|w_t\|^2\right)^{1-\delta_y} + 24G^2T^{\frac{1}{3}}$$

$$+ \frac{C_2\kappa^3 L\gamma}{\lambda(1-\delta_x)G^{2(\delta_x-\delta_y)}}\left(\mathbb{E}\sum_{t=1}^{T}\|v_t\|^2\right)^{1-\delta_x} + \frac{(1-\delta_y)\kappa^2 G^2 C_2}{\lambda^2\mu}(\frac{8\kappa^2 G^2\delta_x(2+\frac{2}{S})}{\lambda^2\mu})^{\frac{\delta_y}{1-\delta_y}}T^{\frac{2\delta_y}{3(1-\delta_y)}}$$

$$+ \frac{24\gamma^2}{1-2\delta_x}(288\gamma^2(2+2S))^{\frac{1-2\delta_x}{2\delta_x}}T^{\frac{1-2\delta_x}{3\delta_x}} + \frac{24\lambda^2}{1-2\delta_y}(\frac{192\lambda^2(2+\frac{2}{S})}{C_2})^{\frac{1-2\delta_y}{2\delta_y}}T^{\frac{1-2\delta_y}{3\delta_y}}.$$

Then we can get

$$\frac{1}{4}\mathbb{E}\sum_{t=1}^{T}(\|\epsilon_t^x\|^2 + \|\nabla_y f(x_t,y_t)\|^2)$$

$$\le \left(\frac{G^{\frac{2}{3}}}{2\lambda} - \frac{\mu}{2}\right)\|y_0 - y_0^*\|^2 + \frac{\lambda L\kappa}{1-\delta_y}\left((2+\frac{2}{S})\mathbb{E}\sum_{t=1}^{T}\|\nabla_y f(x_t,y_t)\|^2\right)^{1-\delta_y} + \frac{24G^2}{C_2}T^{\frac{1}{3}}$$

$$+ \frac{\kappa^3 L\gamma}{\lambda(1-\delta_x)G^{2(\delta_x-\delta_y)}}\left((2+2S)\mathbb{E}\sum_{t=1}^{T}\|\epsilon_t^x\|^2\right)^{1-\delta_x}$$

$$+ \frac{(1-\delta_y)\kappa^2 G^2}{\lambda^2\mu}(\frac{8\kappa^2 G^2\delta_x(2+\frac{2}{S})}{\lambda^2\mu})^{\frac{\delta_y}{1-\delta_y}}T^{\frac{2\delta_y}{3(1-\delta_y)}}$$

$$+ \frac{24\gamma^2}{(1-2\delta_x)C_2}(288\gamma^2(2+2S))^{\frac{1-2\delta_x}{2\delta_x}}T^{\frac{1-2\delta_x}{3\delta_x}} + \frac{24\lambda^2}{(1-2\delta_y)C_2}(\frac{192\lambda^2(2+\frac{2}{S})}{C_2})^{\frac{1-2\delta_y}{2\delta_y}}T^{\frac{1-2\delta_y}{3\delta_y}}.$$

Then we can get

$$\mathbb{E}\sum_{t=1}^{T}(\|\epsilon_t^x\|^2 + \|\nabla_y f(x_t,y_t)\|^2)$$

$$= O(\frac{G^2}{C_2}T^{\frac{1}{3}} + \frac{(\kappa G)^{\frac{2}{1-\delta_y}}}{\mu^{\frac{1}{1-\delta_y}}}T^{\frac{2\delta_y}{3(1-\delta_y)}} + \frac{S^{\frac{1-2\delta_x}{2\delta_x}}}{C_2}T^{\frac{1-2\delta_x}{3\delta_x}} + \frac{1}{C_2^{\frac{1+\delta_y}{3\delta_y}}}T^{\frac{1-2\delta_y}{3\delta_y}}).$$

Moreover, according to Case 2, we can get

$$\mathbb{E}\sum_{t=1}^{T}(\|\nabla_x f(x_t,y_t)\|^2 + \|\nabla_y f(x_t,y_t)\|^2) \le (2+2S)\mathbb{E}\sum_{t=1}^{T}(\|\epsilon_t^x\|^2 + \|\nabla_y f(x_t,y_t)\|^2)$$

$$= O(\frac{G^2 S}{C_2}T^{\frac{1}{3}} + \frac{S(\kappa G)^{\frac{2}{1-\delta_y}}}{\mu^{\frac{1}{1-\delta_y}}}T^{\frac{2\delta_y}{3(1-\delta_y)}} + \frac{S^{\frac{1}{2\delta_x}}}{C_2}T^{\frac{1-2\delta_x}{3\delta_x}} + \frac{S}{C_2^{\frac{1+\delta_y}{3\delta_y}}}T^{\frac{1-2\delta_y}{3\delta_y}}).$$

This complete the proof.

**Case 3:** Assume $\mathbb{E} \sum_{t=1}^{T} \|\nabla_x f(x_t, y_t)\|^2 \geq S\mathbb{E} \sum_{t=1}^{T} \|\epsilon_t^x\|^2$ and $\mathbb{E} \sum_{t=1}^{T} \|\nabla_y f(x_t, y_t)\|^2 \leq S\mathbb{E} \sum_{t=1}^{T} \|\epsilon_t^y\|^2$. Using the condition of this subcase implies

$$\mathbb{E} \sum_{t=1}^{T} \|v_t\|^2 \leq (2 + \frac{2}{S})\mathbb{E} \sum_{t=1}^{T} \|\nabla_x f(x_t, y_t)\|^2,$$

$$\mathbb{E} \sum_{t=1}^{T} \|w_t\|^2 \leq (2 + 2S)\mathbb{E} \sum_{t=1}^{T} \|\epsilon_t^y\|^2.$$

Following Lemma 5 we have:

$$\sum_{t=1}^{T} \|\nabla_x f(x_t, y_t)\|^2$$

$$\leq \sum_{t=1}^{T} \|\epsilon_t^x\|^2 + 4\Phi_*(1/\beta_{T+1})^{\delta_x} (\sum_{t=1}^{T} \|v_t\|^2 + \|w_t\|^2)^{\delta_x} + \frac{L}{1 - 2\delta_x} (\sum_{t=1}^{T} \|v_t\|^2)^{1-2\delta_x} \quad (19)$$

$$\leq \sum_{t=1}^{T} \|\epsilon_t^x\|^2 + \frac{L}{1 - 2\delta_x} (\sum_{t=1}^{T} \|v_t\|^2)^{1-2\delta_x} + 4\Phi_* T^{\frac{2\delta_x}{3}} \Big( \sum_{t=1}^{T} (\|v_t\|^2 + \|w_t\|^2) \Big)^{\delta_x}.$$

Combining Lemma 4 and equation 19 we have:

$$\mathbb{E} \sum_{t=1}^{T} \|\nabla_x f(x_t, y_t)\|^2 + \sum_{t=1}^{T} \|\epsilon_t^y\|^2$$

$$\leq 48G^2 T^{\frac{1}{3}} + \underbrace{\frac{24\gamma^2}{1 - 2\delta_x} T^{\frac{2-4\delta_x}{3}} (\mathbb{E} \sum_{t=1}^{T-1} \|v_t\|^2)^{1-2\delta_x}}_{(a)} + \underbrace{\frac{24\lambda^2}{1 - 2\delta_y} T^{\frac{2-4\delta_y}{3}} (\mathbb{E} \sum_{t=1}^{T-1} \|w_t\|^2)^{1-2\delta_y}}_{(b)} \quad (20)$$

$$+ \sum_{t=1}^{T} \|\epsilon_t^x\|^2 + \frac{L}{1 - 2\delta_x} (\sum_{t=1}^{T} \|v_t\|^2)^{1-2\delta_x} + \underbrace{4\Phi_* T^{\frac{2\delta_x}{3}} \Big( \sum_{t=1}^{T} (\|v_t\|^2 + \|w_t\|^2) \Big)^{\delta_x}}_{(c)}.$$

According to equation 13, setting $\rho = (96\gamma^2(2 + \frac{2}{S}))^{1-2\delta_x}$ for Term (a) we have:

$$a \leq \frac{24\gamma^2}{(1 - 2\delta_x)} (96\gamma^2(2 + \frac{2}{S}))^{\frac{1-2\delta_x}{2\delta_x}} T^{(1-2\delta_x)/3\delta_x} + \frac{1}{4(2 + \frac{2}{S})} \mathbb{E} \sum_{t=1}^{T} \|v_t\|^2. \quad (21)$$

According to equation 13, setting $\rho = (96\lambda^2(2 + 2S))^{1-2\delta_y}$ for Term (b) we have:

$$b \leq \frac{24\lambda^2}{(1 - 2\delta_y)} (96\lambda^2(2 + 2S))^{\frac{1-2\delta_y}{2\delta_y}} T^{(1-2\delta_y)/3\delta_y} + \frac{1}{4(2 + 2S)} \mathbb{E} \sum_{t=1}^{T} \|w_t\|^2. \quad (22)$$

According to equation 15, setting $\rho = (16\delta_x \Phi_*(2 + 2S))^{\delta_x}$ for Term (c) we have:

$$c \leq 4\Phi_*(16\delta_x \Phi_*(2 + 2S))^{\frac{\delta_x}{1-\delta_x}} T^{2\delta_x/3(1-\delta_x)} + \frac{1}{4(2 + 2S)} \mathbb{E} \sum_{t=1}^{T} (\|v_t\|^2 + \|w_t\|^2). \quad (23)$$

Using Case 3, plugging equation 21, equation 22 and equation 23 into equation 20, we have:

$$\frac{5}{12} (\mathbb{E} \sum_{t=1}^{T} \|\nabla_x f(x_t, y_t)\|^2 + \mathbb{E} \sum_{t=1}^{T} \|\epsilon_t^y\|^2)$$

$$\leq 48G^2 T^{\frac{1}{3}} + \frac{L}{1 - 2\delta_x} (\sum_{t=1}^{T} \|v_t\|^2)^{1-2\delta_x} + \frac{24\gamma^2}{(1 - 2\delta_x)} (96\gamma^2(2 + \frac{2}{S}))^{\frac{1-2\delta_x}{2\delta_x}} T^{(1-2\delta_x)/3\delta_x}$$

$$+ \frac{24\lambda^2}{(1 - 2\delta_y)} (96\lambda^2(2 + 2S))^{\frac{1-2\delta_y}{2\delta_y}} T^{(1-2\delta_y)/3\delta_y} + 4\Phi_*(16\delta_x \Phi_*(2 + 2S))^{\frac{\delta_x}{1-\delta_x}} T^{2\delta_x/3(1-\delta_x)}.$$

It implies that:

$$\mathbb{E}\sum_{t=1}^{T}\|\nabla_x f(x_t, y_t)\|^2 + \mathbb{E}\sum_{t=1}^{T}\|\nabla_y f(x_t, y_t)\|^2$$

$$\leq (2 + 2S)\mathbb{E}\sum_{t=1}^{T}\left(\|\epsilon_t^y\|^2 + \|\nabla_x f(x_t, y_t)\|^2\right)$$

$$= O(G^2 S T^{1/3} + S^{2-\frac{1}{2\delta_x}} T^{\frac{1-2\delta_x}{3\delta_x}} + S^{\frac{1}{2\delta_y}} T^{\frac{1-2\delta_y}{3\delta_y}} + S^{\frac{1}{1-\delta_x}} T^{\frac{2\delta_x}{3(1-\delta_x)}}).$$

This complete the proof.

**Case 4:** Assume $\mathbb{E}\sum_{t=1}^{T}\|\nabla_x f(x_t, y_t)\|^2 \geq S\mathbb{E}\sum_{t=1}^{T}\|\epsilon_t^x\|^2$ and $\mathbb{E}\sum_{t=1}^{T}\|\nabla_y f(x_t, y_t)\|^2 \geq S\mathbb{E}\sum_{t=1}^{T}\|\epsilon_t^y\|^2$. Using the condition of this subcase implies

$$\mathbb{E}\sum_{t=1}^{T}\|v_t\|^2 \leq (2 + \frac{2}{S})\mathbb{E}\sum_{t=1}^{T}\|\nabla_x f(x_t, y_t)\|^2,$$

$$\mathbb{E}\sum_{t=1}^{T}\|w_t\|^2 \leq (2 + \frac{2}{S})\mathbb{E}\sum_{t=1}^{T}\|\nabla_y f(x_t, y_t)\|^2.$$

Following Lemma 5 and Lemma 9, letting $C_3 = \min\{\frac{\lambda^2 G^{4(\delta_x - \delta_y)}}{4\kappa^3 L\gamma^2 (2 + \frac{2}{S})}, 1\}$, we have:

$$\sum_{t=1}^{T}\|\nabla_x f(x_t, y_t)\|^2 + C_3\sum_{t=1}^{T}\|\nabla_y f(x_t, y_t)\|^2$$

$$\leq \sum_{t=1}^{T}\|\epsilon_t^x\|^2 + \frac{L}{1 - 2\delta_x}(\sum_{t=1}^{T}\|v_t\|^2)^{1-2\delta_x} + 4\Phi_* T^{\frac{2\delta_x}{3}}\left(\sum_{t=1}^{T}(\|v_t\|^2 + \|w_t\|^2)\right)^{\delta_x}$$

$$+ C_3\left(\frac{L\kappa G^{\frac{2}{3}}}{\lambda} - \mu L\kappa\right)\|y_0 - y_0^*\|^2 + 8\kappa^2 C_3\mathbb{E}\left[\sum_{t=1}^{T}\|\epsilon_t^y\|^2\right] + \frac{C_3\lambda L\kappa}{1 - \delta_y}\mathbb{E}\left(\sum_{t=1}^{T}\|w_t\|^2\right)^{1-\delta_y}$$

$$+ \frac{C_3\kappa^3 L\gamma}{\lambda(1 - \delta_x)G^{2(\delta_x - \delta_y)}}\left(\mathbb{E}\sum_{t=1}^{T}\|v_t\|^2\right)^{1-\delta_x} + \frac{C_3\kappa^3 L\gamma^2}{\lambda^2 G^{4(\delta_x - \delta_y)}}\left(\mathbb{E}\sum_{t=1}^{T}\|v_t\|^2\right)$$

$$+ \frac{C_3\kappa^2 G^2}{\lambda^2\mu}T^{2\delta_y/3}\left(\mathbb{E}\sum_{t=1}^{T}\|w_t\|^2\right)^{\delta_y}.$$

Using Case 4, we can get

$$\frac{11}{16}\sum_{t=1}^{T}\|\nabla_x f(x_t, y_t)\|^2 + \frac{C_3}{2}\sum_{t=1}^{T}\|\nabla_y f(x_t, y_t)\|^2$$

$$\leq \frac{L}{1 - 2\delta_x}(\sum_{t=1}^{T}\|v_t\|^2)^{1-2\delta_x} + \underbrace{4\Phi_* T^{\frac{2\delta_x}{3}}\left(\sum_{t=1}^{T}(\|v_t\|^2 + \|w_t\|^2)\right)^{\delta_x}}_{(d)}$$

$$+ C_3\left(\frac{L\kappa G^{\frac{2}{3}}}{\lambda} - \mu L\kappa\right)\|y_0 - y_0^*\|^2 + \frac{C_3\lambda L\kappa}{1 - \delta_y}\mathbb{E}\left(\sum_{t=1}^{T}\|w_t\|^2\right)^{1-\delta_y} \quad (24)$$

$$+ \frac{C_3\kappa^3 L\gamma}{\lambda(1 - \delta_x)G^{2(\delta_x - \delta_y)}}\left(\mathbb{E}\sum_{t=1}^{T}\|v_t\|^2\right)^{1-\delta_x} + \underbrace{\frac{C_3\kappa^2 G^2}{\lambda^2\mu}T^{2\delta_y/3}\left(\mathbb{E}\sum_{t=1}^{T}\|w_t\|^2\right)^{\delta_y}}_{(e)}.$$

According to equation 15, setting $\rho = (\frac{32(2+\frac{2}{S})\delta_x \Phi_*}{C_3})^{\delta_x}$ for Term (d), then we have

$$e \le 4\Phi_*(1-\delta_x)(\frac{32(2+\frac{2}{S})\delta_x \Phi_*}{C_3})^{\frac{\delta_x}{1-\delta_x}} T^{\frac{2\delta_x}{3(1-\delta_x)}} + \frac{C_3}{8(2+\frac{2}{S})}\mathbb{E}\sum_{t=1}^{T}(\|v_t\|^2 + \|w_t\|^2). \quad (25)$$

Similarly, setting $\rho = (\frac{8(2+\frac{2}{S})\delta_y \kappa^2 G^2}{\lambda^2 \mu})^{\delta_y}$ for Term (e), then we have:

$$e \le \frac{C_3 \kappa^2 G^2}{\lambda^2 \mu}(\frac{8(2+\frac{2}{S})\delta_y \kappa^2 G^2}{\lambda^2 \mu})^{\frac{\delta_y}{1-\delta_y}} T^{\frac{2\delta_y}{3(1-\delta_y)}} + \frac{C_3}{8(2+\frac{2}{S})}\mathbb{E}\sum_{t=1}^{T}\|w_t\|^2. \quad (26)$$

Plugging equation 25 and equation 26 into equation 24, using Case 4 implies:

$$\frac{9}{16}\sum_{t=1}^{T}\|\nabla_x f(x_t, y_t)\|^2 + \frac{C_3}{4}\sum_{t=1}^{T}\|\nabla_y f(x_t, y_t)\|^2$$

$$\le \frac{L}{1-2\delta_x}(\sum_{t=1}^{T}\|v_t\|^2)^{1-2\delta_x} + 4\Phi_*(1-\delta_x)(\frac{32(2+\frac{2}{S})\delta_x \Phi_*}{C_3})^{\frac{\delta_x}{1-\delta_x}} T^{\frac{2\delta_x}{3(1-\delta_x)}}$$

$$+ C_3\left(\frac{L\kappa G^{\frac{2}{3}}}{\lambda} - \mu L\kappa\right)\|y_0 - y_0^*\|^2 + \frac{C_3 \lambda L\kappa}{1-\delta_y}\mathbb{E}\left(\sum_{t=1}^{T}\|w_t\|^2\right)^{1-\delta_y}$$

$$+ \frac{C_3 \kappa^3 L\gamma}{\lambda(1-\delta_x)G^{2(\delta_x - \delta_y)}}\left(\mathbb{E}\sum_{t=1}^{T}\|v_t\|^2\right)^{1-\delta_x} + \frac{C_3 \kappa^2 G^2}{\lambda^2 \mu}(\frac{8(2+\frac{2}{S})\delta_y \kappa^2 G^2}{\lambda^2 \mu})^{\frac{\delta_y}{1-\delta_y}} T^{\frac{2\delta_x}{3(1-\delta_x)}}.$$

It then implies that:

$$\sum_{t=1}^{T}\|\nabla_x f(x_t, y_t)\|^2 + \sum_{t=1}^{T}\|\nabla_y f(x_t, y_t)\|^2$$

$$= O(C_3^{\frac{1}{\delta_x - 1}} T^{\frac{2\delta_x}{3(1-\delta_x)}} + \frac{(\kappa G)^{\frac{2}{1-\delta_y}}}{\mu^{\frac{1}{1-\delta_y}}} T^{\frac{2\delta_x}{3(1-\delta_x)}}).$$

This complete the proof.

Then concluding the above four cases, we can get

$$\sum_{t=1}^{T}\|\nabla_x f(x_t, y_t)\|^2 + \sum_{t=1}^{T}\|\nabla_y f(x_t, y_t)\|^2$$

$$= O(G^2 S T^{\frac{1}{3}} + S^{\frac{1}{2\delta_y}} T^{\frac{1-2\delta_y}{3\delta_y}} + \frac{G^2 S}{C_2} T^{\frac{1}{3}} + \frac{S(\kappa G)^{\frac{2}{1-\delta_y}}}{\mu^{\frac{1}{1-\delta_y}}} T^{\frac{2\delta_y}{3(1-\delta_y)}} + \frac{S^{\frac{1}{2\delta_x}}}{C_2} T^{\frac{1-2\delta_x}{3\delta_x}}$$

$$+ \frac{S}{C_2^{\frac{1+\delta_y}{3\delta_y}}} T^{\frac{1-2\delta_y}{3\delta_y}} + G^2 S T^{1/3} + S^{2-\frac{1}{2\delta_x}} T^{\frac{1-2\delta_x}{3\delta_x}} + S^{\frac{1}{2\delta_y}} T^{\frac{1-2\delta_y}{3\delta_y}} + S^{\frac{1}{1-\delta_x}} T^{\frac{2\delta_x}{3(1-\delta_x)}}$$

$$+ C_3^{\frac{1}{\delta_x - 1}} T^{\frac{2\delta_x}{3(1-\delta_x)}} + \frac{(\kappa G)^{\frac{2}{1-\delta_y}}}{\mu^{\frac{1}{1-\delta_y}}} T^{\frac{2\delta_x}{3(1-\delta_x)}}),$$

where $C_2 = \min\{\frac{\lambda^2 G^{4(\delta_x - \delta_y)}}{12\kappa^3 L\gamma^2(2+2S)}, 1\}$, $C_3 = \min\{\frac{\lambda^2 G^{4(\delta_x - \delta_y)}}{4\kappa^3 L\gamma^2(2+\frac{2}{S})}, 1\}$ and $S \ge 16\kappa^2$. According to $\delta_x > \delta_y$, then we can get

$$\sum_{t=1}^{T} \|\nabla_x f(x_t, y_t)\|^2 + \sum_{t=1}^{T} \|\nabla_y f(x_t, y_t)\|^2$$

$$= O(G^2 S T^{\frac{1}{3}} + S^{\frac{1}{2\delta_y}} T^{\frac{1-2\delta_y}{3\delta_y}} + \frac{G^2 S}{C_2} T^{\frac{1}{3}} + \frac{S(\kappa G)^{\frac{2}{1-\delta_y}}}{\mu^{\frac{1}{1-\delta_y}}} T^{\frac{2\delta_y}{3(1-\delta_y)}} + \frac{S^{\frac{1}{2\delta_x}}}{C_2} T^{\frac{1-2\delta_x}{3\delta_x}}$$

$$+ \frac{S}{C_2^{\frac{1+\delta_y}{3\delta_y}}} T^{\frac{1-2\delta_y}{3\delta_y}} + S^{2-\frac{1}{2\delta_x}} T^{\frac{1-2\delta_x}{3\delta_x}} + S^{\frac{1}{2\delta_y}} T^{\frac{1-2\delta_y}{3\delta_y}} + S^{\frac{1}{1-\delta_x}} T^{\frac{2\delta_x}{3(1-\delta_x)}} + C_3^{\frac{1}{\delta_x - 1}} T^{\frac{2\delta_x}{3(1-\delta_x)}}).$$

Moreover, according to $0.5 > \delta_x > \delta_y$, we can get the following dominant term

$$\sum_{t=1}^{T} \|\nabla_x f(x_t, y_t)\|^2 + \sum_{t=1}^{T} \|\nabla_y f(x_t, y_t)\|^2$$

$$= O\left( \frac{S}{C_2^{\frac{1+\delta_y}{3\delta_y}}} T^{\frac{1-2\delta_y}{3\delta_y}} + S^{\frac{1}{2\delta_y}} T^{\frac{1-2\delta_y}{3\delta_y}} + S^{\frac{1}{1-\delta_x}} T^{\frac{2\delta_x}{3(1-\delta_x)}} + C_3^{\frac{1}{\delta_x - 1}} T^{\frac{2\delta_x}{3(1-\delta_x)}} \right).$$

Then according to the setting of $C_2, C_3$ and $S$, we can get

$$\sum_{t=1}^{T} \|\nabla_x f(x_t, y_t)\|^2 + \sum_{t=1}^{T} \|\nabla_y f(x_t, y_t)\|^2 = O\left( \kappa^{2+\frac{5+5\delta_y}{3\delta_y}} T^{\frac{1-2\delta_y}{3\delta_y}} + \kappa^{\frac{3}{1-\delta_x}} T^{\frac{2\delta_x}{3(1-\delta_x)}} \right).$$

Then setting $\delta_x = \frac{1}{3} + \delta$ and $\delta_y = \frac{1}{3} - \delta$, we can get

$$\sum_{t=1}^{T} \|\nabla_x f(x_t, y_t)\|^2 + \sum_{t=1}^{T} \|\nabla_y f(x_t, y_t)\|^2 \le O\left( \kappa^9 T^{\frac{1}{3}} \right).$$

Utilizing the Cauchy-Schwarz inequality, we can readily derive

$$\frac{1}{T} \left[ \mathbb{E} \sum_{t=1}^{T} \|\nabla_x f(x_t, y_t)\| + \mathbb{E} \sum_{t=1}^{T} \|\nabla_y f(x_t, y_t)\| \right]$$

$$\le \frac{\sqrt{2}}{\sqrt{T}} \left[ \sqrt{\mathbb{E} \sum_{t=1}^{T} \|\nabla_x f(x_t, y_t)\|^2 + \mathbb{E} \sum_{t=1}^{T} \|\nabla_y f(x_t, y_t)\|^2} \right] \le O(\frac{\kappa^{4.5}}{T^{1/3}})$$

This completes the proof. $\qquad\square$

## D  ANALYSIS OF THEOREM 2

In this section, we will replace Assumption 4 with Assumption 5. We present a revised upper bound for $\mathbb{E} \sum_{t=1}^{T} \|\nabla_y f(x_t, y_t)\|^2$, taking into account the $\mu_y$-PL condition.

### D.1  INTERMEDIATE LEMMA OF THEOREM 2

**Lemma 10.** *Under Assumption 1, 2 and 5, we have*

$$\mathbb{E} \sum_{t=1}^{T} \|\nabla_y f(x_t, y_t)\|^2 \le \frac{(16\kappa^2 L^2 + 2\kappa L L_\Phi + \frac{2\kappa L \lambda}{G^{\frac{2}{3}}})\gamma^2}{1 - 2\delta_x} \mathbb{E} \left( \sum_{t=1}^{T} \|v_t\|^2 \right)^{1-2\delta_x}$$

$$+ \frac{2\kappa L^3 \lambda^2}{1 - 2\delta_y} \mathbb{E} \left( \sum_{t=1}^{T} \|w_t\|^2 \right)^{1-2\delta_y} + \frac{4\kappa L \lambda}{G^{2/3}} \mathbb{E} \sum_{t=1}^{T} \|\epsilon_t^y\|^2.$$

*Proof.* Using the smoothness of $f(x, \cdot)$ we have:

$$f(x_{t+1}, y_t) \leq f(x_{t+1}, y_{t+1}) - \eta_t^y \langle \nabla_y f(x_{t+1}, y_t), w_t \rangle + \frac{L}{2} \|y_{t+1} - y_t\|^2.$$

For the term $-\eta_t^y \langle \nabla_y f(x_{t+1}, y_t), w_t \rangle$, we have

$$- \eta_t^y \langle \nabla_y f(x_{t+1}, y_t), w_t \rangle$$

$$\leq -\frac{\eta_t^y}{2} \left( \|\nabla_y f(x_{t+1}, y_t)\|^2 + \|w_t\|^2 - \|\nabla_y f(x_{t+1}, y_t) - \nabla_y f(x_t, y_t) + \nabla_y f(x_t, y_t) - w_t\|^2 \right)$$

$$\leq -\frac{\eta_t^y}{2} \|\nabla_y f(x_{t+1}, y_t)\|^2 - \frac{\eta_t^y}{2} \|w_t\|^2 + \eta_t^y L^2 \|x_{t+1} - x_t\|^2 + \eta_t^y \|\nabla_y f(x_t, y_t) - w_t\|^2$$

$$\leq -\eta_t^y \mu_y \left( \Phi(x_{t+1}) - f(x_{t+1}, y_t) \right) - \frac{\eta_t^y}{2} \|w_t\|^2 + \eta_t^y L^2 \|x_{t+1} - x_t\|^2 + \eta_t^y \|\nabla_y f(x_t, y_t) - w_t\|^2,$$

where the last inequality holds by $\mu_y$-PL condition. Then we have

$$f(x_{t+1}, y_t) \leq f(x_{t+1}, y_{t+1}) - \eta_t^y \mu_y \left( \Phi(x_{t+1}) - f(x_{t+1}, y_t) \right) - \frac{\eta_t^y}{2} \|w_t\|^2$$

$$+ \eta_t^y L^2 \|x_{t+1} - x_t\|^2 + \eta_t^y \|\nabla_y f(x_t, y_t) - w_t\|^2 + \frac{L}{2} \|y_{t+1} - y_t\|^2.$$

Rearranging the above, we have:

$$\Phi(x_{t+1}) - f(x_{t+1}, y_{t+1})$$

$$\leq (1 - \mu_y \eta_t^y) \left( \Phi(x_{t+1}) - f(x_{t+1}, y_t) \right) - \frac{\eta_t^y}{2} \|w_t\|^2 + \eta_t^y L^2 \|x_{t+1} - x_t\|^2 \quad (27)$$

$$+ \eta_t^y \|\nabla_y f(x_t, y_t) - w_t\|^2 + \frac{L}{2} \|y_{t+1} - y_t\|^2.$$

Next, using smoothness of $f(\cdot, y)$, we have:

$$f(x_t, y_t) + \langle \nabla_x f(x_t, y_t), x_{t+1} - x_t \rangle - \frac{L}{2} \|x_{t+1} - x_t\|^2 \leq f(x_{t+1}, y_t).$$

Then we have

$$f(x_t, y_t) - f(x_{t+1}, y_t)$$

$$\leq -\langle \nabla_x f(x_t, y_t), x_{t+1} - x_t \rangle + \frac{L}{2} \|x_{t+1} - x_t\|^2$$

$$= \eta_t^x \langle \nabla_x f(x_t, y_t) - \nabla \Phi(x_t), v_t \rangle - \langle \nabla \Phi(x_t), x_{t+1} - x_t \rangle + \frac{L}{2} \|x_{t+1} - x_t\|^2$$

$$\leq \eta_t^x \omega_t \|\nabla \Phi(x_t) - \nabla_x f(x_t, y_t)\|^2 + \frac{\eta_t^x}{\omega_t} \|v_t\|^2 + \Phi(x_t) - \Phi(x_{t+1}) + \frac{(\eta_t^x)^2 L_\Phi}{2} \|v_t\|^2 + \frac{L(\eta_t^x)^2}{2} \|v_t\|^2$$

$$\leq L^2 \omega_t \eta_t^x \|y_t - y_t^*\|^2 + \frac{\eta_t^x}{\omega_t} \|v_t\|^2 + \Phi(x_t) - \Phi(x_{t+1}) + L_\Phi (\eta_t^x)^2 \|v_t\|^2$$

$$\leq \frac{2L^2 \omega_t \eta_t^x}{\mu_y} \left( \Phi(x_t) - f(x_t, y_t) \right) + \frac{\eta_t^x}{\omega_t} \|v_t\|^2 + \Phi(x_t) - \Phi(x_{t+1}) + L_\Phi (\eta_t^x)^2 \|v_t\|^2,$$

where the second inequality holds by smoothness of $\Phi(x_t)$ and the last two inequality holds by $L < L_\Phi$. The parameter $\omega_t$ will be determined later. Then we have

$$\Phi(x_{t+1}) - f(x_{t+1}, y_t) = \Phi(x_{t+1}) - \Phi(x_t) + \Phi(x_t) - f(x_t, y_t) + f(x_t, y_t) - f(x_{t+1}, y_t)$$

$$\leq (1 + \frac{2L^2 \omega_t \eta_t^x}{\mu_y})(\Phi(x_t) - f(x_t, y_t)) + \frac{\eta_t^x}{\omega_t} \|v_t\|^2 + L_\Phi (\eta_t^x)^2 \|v_t\|^2 \quad (28)$$

Plugging equation 28 into equation 27, we have

$$\Phi(x_{t+1}) - f(x_{t+1}, y_{t+1})$$

$$\leq (1 - \mu_y \eta_t^y)(1 + \frac{2L^2 \omega_t \eta_t^x}{\mu_y})(\Phi(x_t) - f(x_t, y_t)) + \frac{(1 - \mu_y \eta_t^y) \eta_t^x}{\omega_t} \|v_t\|^2$$

$$+ \left( (1 - \mu_y \eta_t^y) L_\Phi + L^2 \eta_t^y \right) (\eta_t^x)^2 \|v_t\|^2 + \left( \frac{L^2 \eta_t^y - 1}{2} \right) \eta_t^y \|w_t\|^2 + \eta_t^y \|\epsilon_t^y\|^2.$$

If $\eta_t^y \geq \frac{1}{\mu}$ for $t = 1, \cdots, t = t_0$, then we have

$$\mathbb{E} \sum_{t=2}^{t_0+1} [(\Phi(x_t) - f(x_t, y_t))]$$

$$\leq \mathbb{E} \sum_{t=1}^{t_0} \frac{\eta_t^y (\eta_t^x)^2 L^2}{2} \|v_t\|^2 + \mathbb{E} \sum_{t=1}^{t_0} \left( \frac{L^2 \eta_t^y - 1}{2} \right) \eta_t^y \|w_t\|^2 + \mathbb{E} \sum_{t=1}^{t_0} \eta_t^y \|\epsilon_t^y\|^2.$$

Now we consider $t = t_0, \cdots, T$. Rearranging the above and summing up, we also have:

$$\mathbb{E} \sum_{t=t_0+1}^{T} \left( \mu \eta_t^y + 2L^2 \omega_t \eta_t^x (\eta_t^y - \frac{1}{\mu}) \right) (\Phi(x_t) - f(x_t, y_t))$$

$$\leq \mathbb{E} \sum_{t=t_0}^{T} (1 - \mu_y \eta_t^y)(\frac{\eta_t^x}{\omega_t} + L_\Phi (\eta_t^x)^2) \|v_t\|^2$$

$$+ \mathbb{E} \sum_{t=t_0}^{T} \frac{\eta_t^y L^2 (\eta_t^x)^2}{2} \|v_t\|^2 + \mathbb{E} \sum_{t=t_0}^{T} \left( \frac{L^2 \eta_t^y - 1}{2} \right) \eta_t^y \|w_t\|^2 + \mathbb{E} \sum_{t=t_0}^{T} \eta_t^y \|\epsilon_t^y\|^2.$$

Setting $\omega_t = \frac{1}{4L^2 \eta_t^x (\frac{1}{\mu} - \eta_t^y)}$, we have $\mu \eta_t^y + 2L^2 \omega_t \eta_t^x (\eta_t^y - \frac{1}{\mu}) \geq \frac{1}{2}$, and $(1 - \mu_y \eta_t^y)(\frac{\eta_t^x}{\omega_t} + L_\Phi (\eta_t^x)^2) \leq (4\kappa L + L_\Phi)(\eta_t^x)^2$ for $t > t_0$. Then we have

$$\frac{1}{2} \mathbb{E} \sum_{t=t_0+1}^{T} [(\Phi(x_t) - f(x_t, y_t))] \leq (4\kappa L + L_\Phi + \frac{L^2 \eta_t^y}{2}) \mathbb{E} \sum_{t=t_0}^{T} (\eta_t^x)^2 \|v_t\|^2$$

$$+ \mathbb{E} \sum_{t=t_0}^{T} \left( \frac{L^2 \eta_t^y - 1}{2} \right) \eta_t^y \|w_t\|^2 + \mathbb{E} \sum_{t=t_0}^{T} \eta_t^y \|\epsilon_t^y\|^2.$$

Summing above two cases, we have

$$\mathbb{E} \sum_{t=1}^{T} [\Phi(x_t) - f(x_t, y_t)]$$

$$\leq (8\kappa L + 2L_\Phi + L^2 \eta_1^y) \mathbb{E} \sum_{t=1}^{T} (\eta_t^x)^2 \|v_t\|^2 + L^2 \mathbb{E} \sum_{t=1}^{T} (\eta_t^y)^2 \|w_t\|^2 + 2\eta_1^y \mathbb{E} \sum_{t=1}^{T} \|\epsilon_t^y\|^2$$

$$\leq \frac{(8\kappa L + 2L_\Phi + \frac{\lambda}{G^{\frac{2}{3}}})\gamma^2}{1 - 2\delta_x} \left( \mathbb{E} \sum_{t=1}^{T} \|v_t\|^2 \right)^{1-2\delta_x} + \frac{L^2 \lambda^2}{1 - 2\delta_y} \left( \mathbb{E} \sum_{t=1}^{T} \|w_t\|^2 \right)^{1-2\delta_y} + \frac{2\lambda}{G^{2/3}} \mathbb{E} \sum_{t=1}^{T} \|\epsilon_t^y\|^2.$$

From Karimi et al. (2016), we know a function is L-smooth and satisfies PL conditions with constant $\mu_y$, it also satisfies the quadratic growth (QG) condition. Using QG we have:

$$\|\nabla_y(x_t, y_t)\|^2 \leq L^2 \|y_t^* - y_t\|^2 \leq 2\kappa L(\Phi(x_t) - f(x_t, y_t)).$$

Then we have

$$\mathbb{E} \sum_{t=1}^{T} \|\nabla_y f(x_t, y_t)\|^2 \leq \frac{(16\kappa^2 L^2 + 2\kappa L L_\Phi + \frac{2\kappa L \lambda}{G^{\frac{2}{3}}})\gamma^2}{1 - 2\delta_x} \left( \mathbb{E} \sum_{t=1}^{T} \|v_t\|^2 \right)^{1-2\delta_x}$$

$$+ \frac{2\kappa L^3 \lambda^2}{1 - 2\delta_y} \left( \mathbb{E} \sum_{t=1}^{T} \|w_t\|^2 \right)^{1-2\delta_y} + \frac{4\kappa L \lambda}{G^{2/3}} \mathbb{E} \sum_{t=1}^{T} \|\epsilon_t^y\|^2.$$

$\square$

### D.2 PROOF OF THEOREM 2

If we change the Assumption from strongly concave to $\mu$-PL condition, this will only affect the upper bound of $\mathbb{E}\sum_{t=1}^{T}\|\nabla_y f(x_t, y_t)\|^2$. We need to reclassify four cases. Introduce constant $P$ and we will give the detailed definition later.

**Case 1:** Assume $\mathbb{E}\sum_{t=1}^{T}\|\nabla_x f(x_t, y_t)\|^2 \leq P\mathbb{E}\sum_{t=1}^{T}\|\epsilon_t^x\|^2$ and $\mathbb{E}\sum_{t=1}^{T}\|\nabla_y f(x_t, y_t)\|^2 \leq P\mathbb{E}\sum_{t=1}^{T}\|\epsilon_t^y\|^2$. Using the condition of this subcase implies

$$\mathbb{E}\sum_{t=1}^{T}(\|v_t\|^2 + w_t\|^2) \leq (2 + 2P)\mathbb{E}\sum_{t=1}^{T}(\|\epsilon_t^x\|^2 + \|\epsilon_t^y\|^2).$$

Similarly, the inequality equation 12 obtained by combining Lemma 3 and 4 does not change when SC is replaced with PL. Then we can get

$$\mathbb{E}\sum_{t=1}^{T}(\|\epsilon_t^x\|^2 + \|\epsilon_t^y\|^2) \leq 96G^2 T^{\frac{1}{3}} + \underbrace{\frac{48\gamma^2}{1 - 2\delta_x} T^{\frac{2-4\delta_x}{3}} (\mathbb{E}\sum_{t=1}^{T-1}\|v_t\|^2)^{1-2\delta_x}}_{\text{(I)}}$$

$$+ \underbrace{\frac{48\lambda^2}{1 - 2\delta_y} T^{\frac{2-4\delta_y}{3}} (\mathbb{E}\sum_{t=1}^{T-1}\|w_t\|^2)^{1-2\delta_y}}_{\text{(II)}} \tag{29}$$

Setting $\rho = (96\gamma^2(2+2P))^{1-2\delta_x}$ for Term (I) and $\rho = (96\lambda^2(2+2P))^{1-2\delta_y}$ for Term (II) we have:

$$\mathbb{E}\sum_{t=1}^{T}(\|\epsilon_t^x\|^2 + \|\epsilon_t^y\|^2)$$

$$\leq 96G^2 T^{\frac{1}{3}} + \frac{1}{2(2+2P)}\mathbb{E}\sum_{t=1}^{T}\|v_t\|^2 + \frac{1}{2(2+2P)}\mathbb{E}\sum_{t=1}^{T}\|w_t\|^2$$

$$+ \frac{96\gamma^2\delta_x}{1 - 2\delta_x}(96\gamma^2(2+2P))^{\frac{1-2\delta_x}{2\delta_x}} T^{\frac{1-2\delta_x}{3\delta_x}} + \frac{96\lambda^2\delta_y}{1 - 2\delta_y}(96\lambda^2(2+2P))^{\frac{1-2\delta_y}{2\delta_y}} T^{\frac{1-2\delta_y}{3\delta_y}}.$$

Denote $P_1 = \max\{\frac{96\gamma^2\delta_x}{1-2\delta_x}(96\gamma^2(2+2P))^{\frac{1-2\delta_x}{2\delta_x}}, \frac{96\lambda^2\delta_y}{1-2\delta_y}(96\lambda^2(2+2P))^{\frac{1-2\delta_y}{2\delta_y}}\}$, according to $1/2 > \delta_x > \delta_y > 0$, we have

$$\mathbb{E}\sum_{t=1}^{T}(\|\epsilon_t^x\|^2 + \|\epsilon_t^y\|^2)$$

$$\leq 96G^2 T^{\frac{1}{3}} + \frac{1}{2(2+2P)}\mathbb{E}\sum_{t=1}^{T}\|v_t\|^2 + \frac{1}{2(2+2P)}\mathbb{E}\sum_{t=1}^{T}\|w_t\|^2 + 2P_1 T^{\frac{1-2\delta_y}{3\delta_y}}.$$

Then we can get:

$$\frac{1}{2}\mathbb{E}\sum_{t=1}^{T}(\|\epsilon_t^x\|^2 + \|\epsilon_t^y\|^2) \leq 96G^2 T^{\frac{1}{3}} + 2P_1 T^{\frac{1-2\delta_y}{3\delta_y}}.$$

Above implies,

$$\mathbb{E}\sum_{t=1}^{T}\|\nabla_x f(x_t, y_t)\|^2 + \mathbb{E}\sum_{t=1}^{T}\|\nabla_y f(x_t, y_t)\|^2$$

$$\leq 2P\mathbb{E}\sum_{t=1}^{T}(\|\epsilon_t^x\|^2 + \|\epsilon_t^y\|^2) = O\left(G^2 P T^{\frac{1}{3}} + P_1 P T^{\frac{1-2\delta_y}{3\delta_y}}\right).$$

Moreover, according to $1/2 > \delta_x > \delta_y > 0$, we have $P_1 = O(P^{\frac{1-2\delta_y}{2\delta_y}})$. Then we can get

$$\mathbb{E}\sum_{t=1}^{T}\|\nabla_x f(x_t, y_t)\|^2 + \mathbb{E}\sum_{t=1}^{T}\|\nabla_y f(x_t, y_t)\|^2 = O(G^2 S T^{\frac{1}{3}} + P^{\frac{1}{2\delta_y}} T^{\frac{1-2\delta_y}{3\delta_y}}).$$

This complete the proof.

**Case 2:** Assume $\mathbb{E}\sum_{t=1}^{T}\|\nabla_x f(x_t, y_t)\|^2 \leq P\mathbb{E}\sum_{t=1}^{T}\|\epsilon_t^x\|^2$ and $\mathbb{E}\sum_{t=1}^{T}\|\nabla_y f(x_t, y_t)\|^2 \geq P\mathbb{E}\sum_{t=1}^{T}\|\epsilon_t^y\|^2$. Using the condition of this subcase implies

$$\mathbb{E}\sum_{t=1}^{T}\|v_t\|^2 \leq (2+2P)\mathbb{E}\sum_{t=1}^{T}\|\epsilon_t^x\|^2, \quad \mathbb{E}\sum_{t=1}^{T}\|w_t\|^2 \leq (2+\frac{2}{P})\mathbb{E}\sum_{t=1}^{T}\|\nabla_y f(x_t, y_t)\|^2.$$

Combining Lemma 3 and Lemma 10 we have

$$\mathbb{E}\sum_{t=1}^{T}\|\epsilon_t^x\|^2 + \mathbb{E}\sum_{t=1}^{T}\|\nabla_y f(x_t, y_t)\|^2 \leq 24G^2 T^{\frac{1}{3}} + \frac{24\gamma^2}{1-2\delta_x}T^{\frac{2-4\delta_x}{3}}(\mathbb{E}\sum_{t=1}^{T-1}\|v_t\|^2)^{1-2\delta_x}$$

$$+ \frac{24\lambda^2}{1-2\delta_y}T^{\frac{2-4\delta_y}{3}}(\mathbb{E}\sum_{t=1}^{T-1}\|w_t\|^2)^{1-2\delta_y} + \frac{(16\kappa^2 L^2 + 2\kappa L L_\Phi + \frac{2\kappa L\lambda}{G^{\frac{2}{3}}})\gamma^2}{1-2\delta_x}\mathbb{E}\left(\sum_{t=1}^{T}\|v_t\|^2\right)^{1-2\delta_x}$$

$$+ \frac{2\kappa L^3 \lambda^2}{1-2\delta_y}\mathbb{E}\left(\sum_{t=1}^{T}\|w_t\|^2\right)^{1-2\delta_y} + \frac{4\kappa L\lambda}{G^{2/3}}\mathbb{E}\sum_{t=1}^{T}\|\epsilon_t^y\|^2.$$

Setting $P \geq \max\{\frac{16\kappa^{20/3}L\lambda}{G^{2/3}}, 4\}$, using Case 2 we can get

$$\mathbb{E}\sum_{t=1}^{T}\|\epsilon_t^x\|^2 + \frac{3}{4}\mathbb{E}\sum_{t=1}^{T}\|\nabla_y f(x_t, y_t)\|^2 \leq 24G^2 T^{\frac{1}{3}} + \underbrace{\frac{24\gamma^2}{1-2\delta_x}T^{\frac{2-4\delta_x}{3}}(\mathbb{E}\sum_{t=1}^{T-1}\|v_t\|^2)^{1-2\delta_x}}_{\text{(III)}}$$

$$+ \underbrace{\frac{24\lambda^2}{1-2\delta_y}T^{\frac{2-4\delta_y}{3}}(\mathbb{E}\sum_{t=1}^{T-1}\|w_t\|^2)^{1-2\delta_y}}_{\text{(IV)}} + \frac{(16\kappa^2 L^2 + 2\kappa L L_\Phi + \frac{2\kappa L\lambda}{G^{\frac{2}{3}}})\gamma^2}{1-2\delta_x}\mathbb{E}\left(\sum_{t=1}^{T}\|v_t\|^2\right)^{1-2\delta_x}$$

$$+ \frac{2\kappa L^3 \lambda^2}{1-2\delta_y}\mathbb{E}\left(\sum_{t=1}^{T}\|w_t\|^2\right)^{1-2\delta_y}.$$

According to equation 13, setting $\rho = (72\gamma^2(2+2P))^{1-2\delta_x}$ for Term (III) we can get

$$\text{III} \leq \frac{24\gamma^2}{1-2\delta_x}(72\gamma^2(2+2P))^{\frac{1-2\delta_x}{2\delta_x}}T^{\frac{1-2\delta_x}{3\delta_x}} + \frac{1}{2(2+2P)}\mathbb{E}\sum_{t=1}^{T}\|v_t\|^2. \tag{30}$$

According to equation 13, setting $\rho = (96\lambda^2(2+\frac{2}{P}))^{1-2\delta_y}$ for Term (IV) we can get

$$\text{IV} \leq \frac{24\lambda^2}{1-2\delta_y}(96\lambda^2(2+\frac{2}{P}))^{\frac{1-2\delta_y}{2\delta_y}}T^{\frac{1-2\delta_y}{3\delta_y}} + \frac{1}{4(2+\frac{2}{P})}\mathbb{E}\sum_{t=1}^{T}\|w_t\|^2. \tag{31}$$

Then we can get

$$
\mathbb{E}\sum_{t=1}^{T}\|\epsilon_t^x\|^2 + \mathbb{E}\sum_{t=1}^{T}\|\nabla_y f(x_t, y_t)\|^2
$$

$$
\leq 24G^2 T^{\frac{1}{3}} + \frac{(16\kappa^2 L^2 + 2\kappa L L_\Phi + \frac{2\kappa L\lambda}{G^{\frac{2}{3}}})\gamma^2}{1 - 2\delta_x}\mathbb{E}\left(\sum_{t=1}^{T}\|v_t\|^2\right)^{1-2\delta_x}
$$

$$
+ \frac{2\kappa L^3\lambda^2}{1 - 2\delta_y}\mathbb{E}\left(\sum_{t=1}^{T}\|w_t\|^2\right)^{1-2\delta_y} + \frac{24\gamma^2}{1 - 2\delta_x}(72\gamma^2(2 + 2P))^{\frac{1-2\delta_x}{2\delta_x}}T^{\frac{1-2\delta_x}{3\delta_x}}
$$

$$
+ \frac{24\lambda^2}{1 - 2\delta_y}(96\lambda^2(2 + \frac{2}{P}))^{\frac{1-2\delta_y}{2\delta_y}}T^{\frac{1-2\delta_y}{3\delta_y}}.
$$

It then follows that

$$
\mathbb{E}\sum_{t=1}^{T}\|\epsilon_t^x\|^2 + \mathbb{E}\sum_{t=1}^{T}\|\nabla_y f(x_t, y_t)\|^2 = O(G^2 T^{\frac{1}{3}} + P^{\frac{1-2\delta_x}{2\delta_x}}T^{\frac{1-2\delta_x}{3\delta_x}} + T^{\frac{1-2\delta_y}{3\delta_y}}).
$$

Moreover, according to Case 2, we can get

$$
\mathbb{E}\sum_{t=1}^{T}\|\nabla_x f(x_t, y_t)\|^2 + \mathbb{E}\sum_{t=1}^{T}\|\nabla_y f(x_t, y_t)\|^2 \leq (2 + 2P)(\mathbb{E}\sum_{t=1}^{T}\|\epsilon_t^x\|^2 + \mathbb{E}\sum_{t=1}^{T}\|\nabla_y f(x_t, y_t)\|^2)
$$

$$
= O(G^2 P T^{\frac{1}{3}} + P^{\frac{1}{2\delta_x}}T^{\frac{1-2\delta_x}{3\delta_x}} + P T^{\frac{1-2\delta_y}{3\delta_y}}).
$$

This complete the proof.

**Case 3:** Assume $\mathbb{E}\sum_{t=1}^{T}\|\nabla_x f(x_t, y_t)\|^2 \geq P\mathbb{E}\sum_{t=1}^{T}\|\epsilon_t^x\|^2$ and $\mathbb{E}\sum_{t=1}^{T}\|\nabla_y f(x_t, y_t)\|^2 \leq P\mathbb{E}\sum_{t=1}^{T}\|\epsilon_t^y\|^2$. Using the condition of this subcase implies

$$
\mathbb{E}\sum_{t=1}^{T}\|v_t\|^2 \leq (2 + \frac{2}{P})\mathbb{E}\sum_{t=1}^{T}\|\nabla_x f(x_t, y_t)\|^2, \quad \mathbb{E}\sum_{t=1}^{T}\|w_t\|^2 \leq (2 + 2P)\mathbb{E}\sum_{t=1}^{T}\|\epsilon_t^y\|^2.
$$

Combinning equation 19 and Lemma 4, using Case 3 we have

$$
\frac{3}{4}\mathbb{E}\sum_{t=1}^{T}\|\nabla_x f(x_t, y_t)\|^2 + \mathbb{E}\sum_{t=1}^{T}\|\epsilon_t^y\|^2
$$

$$
\leq \frac{L}{1 - 2\delta_x}(\sum_{t=1}^{T}\|v_t\|^2)^{1-2\delta_x} + \underbrace{4\Phi_* T^{\frac{2\delta_x}{3}}\left(\sum_{t=1}^{T}(\|v_t\|^2 + \|w_t\|^2)\right)^{\delta_x} + 48G^2 T^{\frac{1}{3}}}_{\text{(a)}}
$$

$$
+ \underbrace{\frac{24\gamma^2}{1 - 2\delta_x}T^{\frac{2-4\delta_x}{3}}(\mathbb{E}\sum_{t=1}^{T-1}\|v_t\|^2)^{1-2\delta_x}}_{\text{(b)}} + \underbrace{\frac{24\lambda^2}{1 - 2\delta_y}T^{\frac{2-4\delta_y}{3}}(\mathbb{E}\sum_{t=1}^{T-1}\|w_t\|^2)^{1-2\delta_y}}_{\text{(c)}}.
$$

According to equation 15, setting $\rho = (16\Phi_*\delta_x(2 + 2P))^{\delta_x}$ for Term (a), we have

$$
\text{a} \leq 4\Phi_*(16\Phi_*\delta_x(2 + 2P))^{\frac{\delta_x}{1-\delta_x}}T^{\frac{2\delta_x}{3(1-\delta_x)}} + \frac{1}{4(2 + 2P)}\mathbb{E}\sum_{t=1}^{T}(\|v_t\|^2 + \|w_t\|^2).
$$

According to equation 13, setting $\rho = (96\gamma^2(2 + \frac{2}{P}))^{1-2\delta_x}$ for Term (b) we have

$$
\text{b} \leq \frac{24\gamma^2}{1 - 2\delta_x}(96\gamma^2(2 + \frac{2}{P}))^{\frac{1-2\delta_x}{2\delta_x}}T^{\frac{1-2\delta_x}{3\delta_x}} + \frac{1}{4(2 + \frac{2}{P})}\mathbb{E}\sum_{t=1}^{T}\|v_t\|^2.
$$

According to equation 13, setting $\rho = (96\lambda^2(2+2P))^{1-2\delta_y}$ for Term (c) we have

$$c \leq \frac{24\lambda^2}{1-2\delta_y}(96\lambda^2(2+2P))^{\frac{1-2\delta_y}{2\delta_y}}T^{\frac{1-2\delta_y}{3\delta_y}} + \frac{1}{4(2+2P)}\mathbb{E}\sum_{t=1}^{T}\|w_t\|^2.$$

Then we can conclude

$$\frac{1}{4}\sum_{t=1}^{T}\|\nabla_x f(x_t, y_t)\|^2 + \frac{1}{4}\mathbb{E}\sum_{t=1}^{T}\|\epsilon_t^y\|^2$$

$$\leq \frac{L}{1-2\delta_x}(\sum_{t=1}^{T}\|v_t\|^2)^{1-2\delta_x} + 48G^2 T^{\frac{1}{3}} + 4\Phi_*(16\Phi_*\delta_x(2+2P))^{\frac{\delta_x}{1-\delta_x}}T^{\frac{2\delta_x}{3(1-\delta_x)}}$$

$$+ \frac{24\gamma^2}{1-2\delta_x}(96\gamma^2(2+\frac{2}{P}))^{\frac{1-2\delta_x}{2\delta_x}}T^{\frac{1-2\delta_x}{3\delta_x}} + \frac{24\lambda^2}{1-2\delta_y}(96\lambda^2(2+2P))^{\frac{1-2\delta_y}{2\delta_y}}T^{\frac{1-2\delta_y}{3\delta_y}}.$$

It implies that:

$$\sum_{t=1}^{T}\|\nabla_x f(x_t, y_t)\|^2 + \mathbb{E}\sum_{t=1}^{T}\|\epsilon_t^y\|^2 = O(G^2 T^{\frac{1}{3}} + P^{\frac{\delta_x}{1-\delta_x}}T^{\frac{2\delta_x}{3(1-\delta_x)}} + T^{\frac{1-2\delta_x}{3\delta_x}} + P^{\frac{1-2\delta_y}{2\delta_y}}T^{\frac{1-2\delta_y}{3\delta_y}}).$$

Then according to Case 3, we can get

$$\mathbb{E}\sum_{t=1}^{T}\|\nabla_x f(x_t, y_t)\|^2 + \mathbb{E}\sum_{t=1}^{T}\|\nabla_y f(x_t, y_t)\|^2 \leq (2+2P)(\sum_{t=1}^{T}\|\nabla_x f(x_t, y_t)\|^2 + \mathbb{E}\sum_{t=1}^{T}\|\epsilon_t^y\|^2)$$

$$= O(G^2 P T^{\frac{1}{3}} + P^{\frac{1}{1-\delta_x}}T^{\frac{2\delta_x}{3(1-\delta_x)}} + PT^{\frac{1-2\delta_x}{3\delta_x}} + P^{\frac{1}{2\delta_y}}T^{\frac{1-2\delta_y}{3\delta_y}}).$$

This complete the proof.

**Case 4:** Assume $\mathbb{E}\sum_{t=1}^{T}\|\nabla_x f(x_t, y_t)\|^2 \geq P\mathbb{E}\sum_{t=1}^{T}\|\epsilon_t^x\|^2$ and $\mathbb{E}\sum_{t=1}^{T}\|\nabla_y f(x_t, y_t)\|^2 \geq P\mathbb{E}\sum_{t=1}^{T}\|\epsilon_t^y\|^2$. Using the condition of this subcase implies

$$\mathbb{E}\sum_{t=1}^{T}\|v_t\|^2 \leq (2+\frac{2}{P})\mathbb{E}\sum_{t=1}^{T}\|\nabla_x f(x_t, y_t)\|^2,$$

$$\mathbb{E}\sum_{t=1}^{T}\|w_t\|^2 \leq (2+\frac{2}{P})\mathbb{E}\sum_{t=1}^{T}\|\nabla_y f(x_t, y_t)\|^2.$$

Following Lemma 5 and Lemma 10, we have:

$$\mathbb{E}\sum_{t=1}^{T}\|\nabla_x f(x_t, y_t)\|^2 + \mathbb{E}\sum_{t=1}^{T}\|\nabla_y f(x_t, y_t)\|^2$$

$$\leq \sum_{t=1}^{T}\|\epsilon_t^x\|^2 + \frac{L}{1-2\delta_x}(\sum_{t=1}^{T}\|v_t\|^2)^{1-2\delta_x} + 4\Phi_* T^{\frac{2\delta_x}{3}}\left(\sum_{t=1}^{T}(\|v_t\|^2 + \|w_t\|^2)\right)^{\delta_x}$$

$$+ \frac{(16\kappa^2 L^2 + 2\kappa L L_\Phi + \frac{2\kappa L\lambda}{G^{\frac{2}{3}}})\gamma^2}{1-2\delta_x}\mathbb{E}\left(\sum_{t=1}^{T}\|v_t\|^2\right)^{1-2\delta_x} + \frac{2\kappa L^3\lambda^2}{1-2\delta_y}\mathbb{E}\left(\sum_{t=1}^{T}\|w_t\|^2\right)^{1-2\delta_y}$$

$$+ \frac{4\kappa L\lambda}{G^{2/3}}\mathbb{E}\sum_{t=1}^{T}\|\epsilon_t^y\|^2.$$

According to Case 4, we can get

$$\frac{3}{4}(\mathbb{E}\sum_{t=1}^{T}\|\nabla_x f(x_t,y_t)\|^2 + \mathbb{E}\sum_{t=1}^{T}\|\nabla_y f(x_t,y_t)\|^2)$$

$$\leq \frac{L}{1-2\delta_x}(\sum_{t=1}^{T}\|v_t\|^2)^{1-2\delta_x} + \underbrace{4\Phi_* T^{\frac{2\delta_x}{3}}\Big(\sum_{t=1}^{T}(\|v_t\|^2 + \|w_t\|^2)\Big)^{\delta_x}}_{(d)}$$

$$+ \frac{(16\kappa^2 L^2 + 2\kappa LL_\Phi + \frac{2\kappa L\lambda}{G^{\frac{2}{3}}})\gamma^2}{1-2\delta_x}\mathbb{E}\left(\sum_{t=1}^{T}\|v_t\|^2\right)^{1-2\delta_x} + \frac{2\kappa L^3\lambda^2}{1-2\delta_y}\mathbb{E}\left(\sum_{t=1}^{T}\|w_t\|^2\right)^{1-2\delta_y}.$$

According to equation 15, setting $\rho = (16\delta_x\Phi_*(2+\frac{2}{P}))^{\delta_x}$, we can get

$$d \leq 4\Phi_*(16\delta_x\Phi_*(2+\frac{2}{P}))^{\frac{\delta_x}{1-\delta_x}}T^{\frac{2\delta_x}{3(1-\delta_x)}} + \frac{1}{4(2+\frac{2}{P})}\mathbb{E}\sum_{t=1}^{T}(\|v_t\|^2 + \|w_t\|^2),$$

Then we can get

$$\frac{1}{2}(\mathbb{E}\sum_{t=1}^{T}\|\nabla_x f(x_t,y_t)\|^2 + \mathbb{E}\sum_{t=1}^{T}\|\nabla_y f(x_t,y_t)\|^2)$$

$$\leq \frac{L}{1-2\delta_x}(\sum_{t=1}^{T}\|v_t\|^2)^{1-2\delta_x} + \frac{(16\kappa^2 L^2 + 2\kappa LL_\Phi + \frac{2\kappa L\lambda}{G^{\frac{2}{3}}})\gamma^2}{1-2\delta_x}\mathbb{E}\left(\sum_{t=1}^{T}\|v_t\|^2\right)^{1-2\delta_x}$$

$$+ \frac{2\kappa L^3\lambda^2}{1-2\delta_y}\mathbb{E}\left(\sum_{t=1}^{T}\|w_t\|^2\right)^{1-2\delta_y} + 4\Phi_*(16\delta_x\Phi_*(2+\frac{2}{P}))^{\frac{\delta_x}{1-\delta_x}}T^{\frac{2\delta_x}{3(1-\delta_x)}}.$$

It implies that:

$$\left[\mathbb{E}\sum_{t=1}^{T}\|\nabla_x f(x_t,y_t)\|^2 + \mathbb{E}\sum_{t=1}^{T}\|\nabla_y f(x_t,y_t)\|^2\right] = O(T^{\frac{2\delta_x}{3(1-\delta_x)}}).$$

Then we conclude above four cases. We can get

$$\mathbb{E}\sum_{t=1}^{T}\|\nabla_x f(x_t,y_t)\|^2 + \mathbb{E}\sum_{t=1}^{T}\|\nabla_y f(x_t,y_t)\|^2$$

$$= O(G^2 PT^{\frac{1}{3}} + P_1 PT^{\frac{1-2\delta_y}{3\delta_y}} + G^2 PT^{\frac{1}{3}} + P^{\frac{1}{2\delta_x}}T^{\frac{1-2\delta_x}{3\delta_x}} + PT^{\frac{1-2\delta_y}{3\delta_y}}$$

$$+ G^2 PT^{\frac{1}{3}} + P^{\frac{1}{1-\delta_x}}T^{\frac{2\delta_x}{3(1-\delta_x)}} + PT^{\frac{1-2\delta_x}{3\delta_x}} + P^{\frac{1}{2\delta_y}}T^{\frac{1-2\delta_y}{3\delta_y}} + T^{\frac{2\delta_x}{3(1-\delta_x)}}),$$

where $P_1 = \max\{\frac{96\gamma^2\delta_x}{1-2\delta_x}(96\gamma^2(2+2P))^{\frac{1-2\delta_x}{2\delta_x}}, \frac{96\lambda^2\delta_y}{1-2\delta_y}(96\lambda^2(2+2P))^{\frac{1-2\delta_y}{2\delta_y}}\}$, $P \geq \max\{\frac{16\kappa^{20/3}L\lambda}{G^{2/3}}, 4\}$ and $\frac{1}{2} > \delta_x > \delta_y$. Then we can get the following dominant term

$$\mathbb{E}\sum_{t=1}^{T}\|\nabla_x f(x_t,y_t)\|^2 + \mathbb{E}\sum_{t=1}^{T}\|\nabla_y f(x_t,y_t)\|^2$$

$$= O(P_1 PT^{\frac{1-2\delta_y}{3\delta_y}} + P^{\frac{1}{1-\delta_x}}T^{\frac{2\delta_x}{3(1-\delta_x)}} + P^{\frac{1}{2\delta_y}}T^{\frac{1-2\delta_y}{3\delta_y}}).$$

Then it follows that

$$\mathbb{E}\sum_{t=1}^{T}\|\nabla_x f(x_t,y_t)\|^2 + \mathbb{E}\sum_{t=1}^{T}\|\nabla_y f(x_t,y_t)\|^2 = O\left(\kappa^{\frac{20}{3(1-\delta_x)}}T^{\frac{2\delta_x}{3(1-\delta_x)}} + \kappa^{\frac{10}{3\delta_y}}T^{\frac{1-2\delta_y}{3\delta_y}}\right).$$

setting $\delta_x = \frac{1}{3} + \delta$ and $\delta_y = \frac{1}{3} - \delta$, we can get

$$\mathbb{E} \sum_{t=1}^{T} \|\nabla_x f(x_t, y_t)\|^2 + \mathbb{E} \sum_{t=1}^{T} \|\nabla_y f(x_t, y_t)\|^2 \leq O(\kappa^{10} T^{\frac{1}{3}}).$$

Utilizing the Cauchy-Schwarz inequality, we can readily derive

$$\frac{1}{T} \left[ \mathbb{E} \sum_{t=1}^{T} \|\nabla_x f(x_t, y_t)\| + \mathbb{E} \sum_{t=1}^{T} \|\nabla_y f(x_t, y_t)\| \right]$$

$$\leq \frac{\sqrt{2}}{\sqrt{T}} \left[ \sqrt{\mathbb{E} \sum_{t=1}^{T} \|\nabla_x f(x_t, y_t)\|^2 + \mathbb{E} \sum_{t=1}^{T} \|\nabla_y f(x_t, y_t)\|^2} \right] \leq O(\frac{\kappa^5}{T^{1/3}}).$$

This completes the proof. $\square$

