# OpenReview forum: "AdaFM: Adaptive Variance-Reduced Algorithm for Stochastic Minimax Optimization"
_ICLR.cc/2025/Conference — Submitted to ICLR 2025_

### Official Review · Reviewer_ut2G · 2024-11-03

**Soundness:** 3
**Presentation:** 2
**Contribution:** 2
**Rating:** 3
**Confidence:** 4

**Summary:**

*Summary of the paper:*

 The paper investigates a setting of minimax zero-sum game with non-convex-strongly-concave structure.
The authors assume a smooth minimax objective with bounded gradients, and introduce a variance reduction technique combined with adaptive stepsize to obtain convergence rates for the problem.






*Summary:*

 I think that the setting of the paper is a bit lacking or weak regarding the assumptions that are made, that the methods is not properly compared to previous approach,  and that the adaptivity that is claimed is not very beneficial.

**Strengths:**

-The paper does provide convergence guarantees, for the aforementioned problem, with an adaptive learning rate which is different from the one in the related VARAdaGDA

-The fact that the learning rate for the x depends on $\alpha_t^x$ and $\alpha_t^y$ is interesting.

**Weaknesses:**

-The authors consider a crude assumption that the gradients are bounded, compared to a more refined assumption of bounded variance which is much more standard in the literature.

-The authors seem to have missed an important point about adaptivity: which is adaptivity to the noise variance. Concretely, the complex learning rate in the STORM paper is designed to ensure that in the case where the variance is zero, then STORM guarantees the faster convergence rate known for the noiseless case.  This does not apply to the current paper, and also cannot apply since they use a non-adaptive momentum of $1/t^{2/3}$, which can only provide rates that optimally match the noisy case.

-The authors do not compare the dependence of their convergence rate to the one of VARAdaGDA, in terms of $G,L$, it seems that the fact that they can take the constants $\gamma,\lambda$ to be $1$, may substantially degrade the performance with respect to $G,L$. This again means that the claimed adaptivity is not really beneficial.

-In term of proof/algorithmic techniques, I do not think that there is anything very novel or challenging.

**Questions:**

What is the benefit of adaptivity in your work? Currently, this is not clear to me

---

### Official Review · Reviewer_ukjW · 2024-11-03

**Soundness:** 2
**Presentation:** 3
**Contribution:** 2
**Rating:** 3
**Confidence:** 4

**Summary:**

This paper provides an algorithm for solving minimax problems using variance reduction in an adaptive manner.

The basic idea appears to use a sequence of learning rates loosely inspired by previous adaptive methods (e.g. STORM).

**Strengths:**

The problem is interesting, and so far as I know is not solved in the literature. The overall approach seems promising.

**Weaknesses:**

I do not understand what exactly is adaptive about the proposed method.

More specifically, “adaptivity” in optimization typically means that there is some problem parameter (e.g. variance in gradients, smoothness constant, strong convexity constant etc) such that without prior knowledge of the parameter, the algorithm is able to achieve nearly the same convergence guarantee as would be possible given prior knowledge of the parameter. Usually this means that even if the algorithm has some hyperparameters, these parameters have minimal impact on the convergence guarantee. I don’t see how that occurs here.

The authors appear to not be considering the variance of the gradients (Assumption 2 is just a coarse Lipschitz bound), so the only parameter to “adapt” to is the condition number $\kappa$. However, the convergence guarantees do not explain the impact of the algorithm hyperparameters at all - it is all hidden in the big-Oh. Moreover, it appears that perhaps the dependence on $\kappa$ is not even correct: https://arxiv.org/pdf/2308.09604 appears to achieve a much better complexity of $\kappa^3/\epsilon^3$ in a non-adaptive way. This work appears to obtain $\kappa^{15}/\epsilon^3$ instead. I’m willing to believe that proper setting of $\lambda$ or $\gamma$ would alleviate this issue, but that is not obvious since the influence of these parameters was not provided in the theorem statement. Moreover, this would defeat the purpose of the adaptivity.

In summary, I don’t think this paper accomplishes the stated goal of adaptivity. Perhaps the authors meant a different goal, but if so they should clearly explain what they are trying to achieve and why it is desirable.

**Questions:**

Is there a way in which the algorithm is adaptive? If so, what exactly is the claim? What are the dependencies on the parameters?

---

### Official Review · Reviewer_EU8u · 2024-11-04

**Soundness:** 3
**Presentation:** 3
**Contribution:** 3
**Rating:** 8
**Confidence:** 3

**Summary:**

This paper proposes a new adaptive variance-reduced algorithm for minimax optimization. By using STORM as the base optimizer with carefully designed learning rates, the algorithm converges to the optimal rate without the need for manually tuning the hyperparameters.

**Strengths:**

- The problem is well-motivated. Modern ML algorithms usually require a lot of hyperparameter tuning to achieve good performance, which could be very computationally expensive for deep models with large datasets. Thus, there's a huge for developing adaptive and robust algorithms that do not require too much tuning.

- The algorithm is simple and achieved due to some cool and clever settings of the learning rate. The proposed method also achieved an almost optimal convergence rate.

- The author also provides empirical validation of their proposed method. AdaFM seems to be quite competitive with the best current method in multiple settings.

- The paper is well-written overall.

**Weaknesses:**

- The results seem to only apply for SC and PL settings (since in the concave case, $\mu$ is zero - thus blowing up the final bound), which limits the application of the method.

**Questions:**

- How should we interpret the results shown in Figure 5? Do the results show that AdaFM is more robust to a wider range of learning rate settings?

- Do the authors have any possible explanation for the initial increase in the gradient norm in Figure 2c.

- Can we use the same formula for the learning rate to derive a parameter-free version of STORM? As far as the reviewer knows, the original STORM is not parameter-free.

---

### Official Review · Reviewer_Byeu · 2024-11-11

**Soundness:** 3
**Presentation:** 3
**Contribution:** 3
**Rating:** 6
**Confidence:** 2

**Summary:**

This paper presents an adaptive filtered momentum method that achieves near optimal convergence rates for non-convex strongly concave and non-convex PL settings for stochastic minimax optimization; a particular advantage of the method is that it doesn't appear to require many tuning parameters which makes it particularly attractive for practical optimization problems.

**Strengths:**

The paper is relatively well written and presents what appears to be a novel algorithm for solving stochastic minmax optimization problems. I am not an expert in this area, so I cannot comment on the full degree of novelty of this paper compared to existing works.

**Weaknesses:**

None.

**Questions:**

None.

---

### Meta-Review · Area_Chair_4yxk · 2024-12-09

**Metareview:**

The paper introduces Adaptive Filtered Momentum (AdaFM), an adaptive variance-reduction algorithm for stochastic minimax optimization problems. AdaFM combines variance reduction with an adaptive stepsize to achieve convergence guarantees for non-convex-strongly-concave and non-convex-Polyak-Łojasiewicz objectives. The authors claim that their method eliminates the need for manual hyperparameter tuning by adjusting momentum and learning rates based on historical estimator information.

Reviewers expressed concerns about the assumptions made (bounded gradients instead of bounded variance), the lack of comparison to existing methods like VARAdaGDA, and the claimed adaptivity of the algorithm. The paper does not adequately demonstrate how AdaFM adapts to noise variance or how its performance compares to other methods, particularly in terms of key parameters. The non-adaptive momentum also limits the claimed benefits of adaptivity.

**Additional Comments On Reviewer Discussion:**

The authors did not address the reviewers' questions.

---

### Decision · Program_Chairs · 2025-01-22

Reject